



# An improved OMI ozone profile research product version 2.0 with collection 4 L1b data and algorithm updates

Juseon Bak[1,*], Xiong Liu[2], Yang Kai[3], Gonzalo Gonzalez Abad[2],

Ewan O`Sullivan[2], Kelly Chance[2], Cheol-Hee Kim[1,4]

[1]Institute of Environmental Studies, Pusan National University, Busan 46241, Republic of Korea

[2]Smithsonian Astrophysical Observatory (SAO), Center for Astrophysics | Harvard & Smithsonian, Cambridge, MA 02138, USA

[3]Department of Atmospheric Sciences, University of Maryland, College Park, MD 20742, USA.

[4]Department of Atmospheric Sciences, Pusan National University, Busan 46241, South Korea

*Corresponding Author Juseon Bak (juseonbak@pusan.ac.kr) & Xiong Liu (xliu@cfa.harvard.edu)

## Abstract

We describe the new and improved version (V2) of the ozone profile research product from the Ozone Monitoring Instrument (OMI) on the Aura satellite. One of the major changes is to switch the OMI L1b data from collection 3 to the recent collection 4 as well as the accompanying auxiliary datasets. The algorithm details are updated on radiative transfer (RT) model calculation and measurement calibrations, along with the input changes of meteorological data, and with the use of a tropopause-based ozone profile climatology, an improved high-resolution solar reference spectrum, and a recent ozone absorption cross-section dataset. A super Gaussian is applied to better represent OMI slit functions, instead of a normal Gaussian. The effect of slit function errors on the spectral residuals is further accounted for as pseudo absorbers in the iterative fit process. The OMI irradiances are averaged into monthly composites to reduce noise uncertainties in OMI daily measurements and to cancel out the common degradation of radiance and irradiance measurements which was previously neglected due to use of climatological composites. The empirical soft calibration spectra are re-derived to be consistent with the updated implementations and derived annually to remove the time-dependent systematic biases between measured and simulated radiances. The "common mode" correction spectra are derived from remaining residual spectra after soft calibration as a function of solar zenith angle. The common mode is included as a pseudo absorber in the iterative fit process, which helps to reduce the discrepancies of ozone retrieval accuracy between lower and higher solar zenith angles and between nadir and off-nadir pixels. Validation with ozonesonde measurements demonstrates the improvements of ozone profile retrievals in the troposphere, especially around the tropopause. The retrieval quality of tropospheric column ozone is improved with respect to the seasonal consistency between winter and summer as well as the long-term consistency before and after the row-anomaly occurrence.



# 1. Introduction

The Smithsonian Astrophysical Observatory (SAO) ozone profile algorithm was originally developed to retrieve ozone profiles with sensitivity down to the lower troposphere from Global Ozone Monitoring Experiment (GOME) measurements (Liu et al., 2005) and has been continuously adapted to Ozone Monitoring Instrument (OMI) (Liu et al., 2010), GOME/2A (Cai et al., 2012), Ozone Mapping and Profiler Suite (OMPS) (Bak et al., 2017), TROPOspheric Monitoring Instrument (TROPOMI) (Zhao et al., 2021), Geostationary Environment Monitoring Spectrometer (GEMS) (Bak et al., 2019a), and Tropospheric Emissions: Monitoring of Pollution (TEMPO) (Zoogman et al., 2017). The SAO algorithm has been put into production in the NASA's OMI Science Investigator-led Processing System (SIPS) to create the OMI ozone profile research product titled OMPROFOZ v0.93 that is publically distributed via the Aura Validation Data Center (AVDC) (https://avdc.gsfc.nasa.gov/pub/data/satellite/Aura/OMI/V03/L2/OMPROFOZ/). The OMPROFOZ product has been contributed to a better understanding of chemical and dynamical ozone variability associated with anthropogenic pollution over central and eastern China (Hayashida et al., 2015; Wei et al., 2022), transport of anthropogenic pollution in free troposphere (Walker et al., 2010) and stratospheric ozone intrusion (Kuang et al., 2017) as well as ozone concentration changes in the Asian summer monsoon (Lu et al., 2018; Luo et al., 2018). Furthermore, this product has been used to quantify the global tropospheric budget of ozone and evaluate how well current chemistry-climate models reproduce the observations (Hu et al., 2017; Zhang et al., 2010). Compared to other similar space-borne UV instruments, OMI has maintained much better long-term stability over the mission with low optical degradation (1-2 % in radiance, 3-8 % in irradiance) and high wavelength stability (0.005-0.020 nm), but there has been concern over the row anomaly effects appearing in 2007, becoming serious in early 2009, and currently damaging about half of the instrument's viewing capability (Schenkeveld et al., 2017). So far, satellite ozone profile products have not been reliable for long-term analysis, especially for the tropospheric ozone measurements due to their susceptibility to the optical degradation of instruments (Gaudel et al., 2018). Ten-years of the OMPROFOZ product were assessed in-depth in Huang et al. (2018;2017) through the spatiotemporal validation using global reference dataset collected from in-situ ozonesonde and space-borne Microwave Limb Sounder (MLS) measurements. They concluded noticeable discrepancies in time-series of data quality due to the occurrence of serious row anomaly and the dependence of retrieval quality on the latitude/season/viewing geometries. Since the first release of OMPROFOZ data, implementation details have been externally refined to improve the retrieval quality. Bak et al., (2013) demonstrated improvements of ozone profile retrievals around the extratropical tropopause area by better constraining climatological a priori information. To better



represent an instrument spectral response function (ISRF), Sun et al. (2017) employed a Super Gaussian
function which can represent more complex shapes compared to a classical Gaussian function. The slit
function linearization was experimented in Bak et al. (2019b) to account for the effects of errors in slit
function parameters on the spectral fit residuals. Moreover, the best spectroscopic inputs were
investigated with respect to the ozone cross-section (Bak et al., 2020; Liu et al., 2013) and the high-
resolution solar reference spectrum (Bak et al., 2022). To accelerate the time-consuming radiative
transfer calculation, a principal component analysis (PCA)-based radiative transfer (RT) model was
employed as a forward model with the correction scheme of RT approximation errors using look-up
tables (LUTs) (Bak et al., 2021). The updates to radiometric corrections were made with the time-
dependent soft calibration and solar zenith angle dependent common mode correction, improving the
spatiotemporal consistency of retrieval quality, which are detailed in this paper. Individual refinements
mentioned above are incorporated in the OMPROFOZ V2 algorithm, along with the switch of OMI L1b
data product from collection 3 to collection 4. Note that OMI measurements have been reprocessed to
deliver the new collection 4 dataset which supersedes and improves the collection 3 with respect to the
ongoing instrument effects and optical degradations, drifts in electronic gain, and pixel quality flagging.
(Kleipool et al., 2022).
In this paper we describe updates made in the OMI ozone profile algorithm, discuss their impact on
spectral fit and ozone profile retrievals, and provide an initial quantitative assessment of tropospheric
ozone columns with respect to their long-term consistency. Section 2 describes OMI L1b and auxiliary
products used in retrieving ozone profiles, along with the retrieval methodology and OMPROFOZ v2
product. In section 3 the updates of implementation details are specified and verified. Section 4 presents
the validation results using ozonesonde measurements. This paper is summarized and concluded in
Section 5.

## 2. Description of the SAO OMI ozone profile algorithm and OMPROFOZ product

### 2.1 OMI products

Table 1 lists the OMI standard or auxiliary products used in reprocessing OMI ozone profiles,
which are publicly available through NASA's Goddard Earth Sciences Data and Information Services
Center (GES DISC). OMI is a nadir-viewing UV and visible spectrometer in which two-dimensional
(spectral × spatial) charged-coupled device (CCD) detectors are employed. The collection 4 L0-1B
processor was newly built based on the TROPOMI L0-1B processor at the OMI SIPS, which produces





radiometrically calibrated and geolocated solar irradiances and earthshine radiances from the raw sensor measurements. Insights learned from the usage of OMI collection 3 data over the past 17 years are leveraged to correct optical and electronic aging and improve pixel quality flagging. The details of updates and improvements from collection 3 to 4 can be found in Kleipool et al. (2022). The OML1BIRR (10.5067/Aura/OMI/DATA1401) provides the daily averaged irradiance measurements. The OML1BRUG (10.5067/AURA/OMI/DATA1402) contains Earth view spectral radiances taken in the global mode from the UV detector. To increase a signal to noise ratio (SNR) at shorter UV wavelengths, a measured spectrum is divided into two sub channels at ~ 310 nm and then the spatial resolution of the shorter spectra is degraded by a factor of 2 in cross-track pixels, resulting into 48 km and 24 km at nadir for 159 channels in the Band 1 (UV-1, 264-311 nm) and for 557 channels in the Band 2 (UV-2, 307-383 nm), respectively. The spatial resolution is 13 km in the flight direction. Cloud information is taken from OMCLDO2 based on the spectral fitting of $O_2$-$O_2$ absorption band at 477 nm, while a climatological surface albedo is taken from OMLER. The OMUANC is a new ancillary product geo-collocated to UV2 spatial pixels, developed for supporting the production of the OMI L2 data products. We use OMUANC data for taking snow ice flags and row anomaly flags. The row anomaly is an anomaly which affects the quality of the level 1B radiance data at all wavelengths for specific viewing angles. Only two of OMI's 60 rows were initially affected in 2007, but the anomalies have become more serious since January 2009 (~ 30%), spreading to ~ 50 % (rows 25-55) during the period of 2010-2012. There is no reliable correction scheme for the row anomaly-affected measurements and therefore flagging the row anomalies as bad data is important to assure the L2 product quality. Row anomaly flags are available from both OML1BRUG and OMUANC; the former is based on analysis of features observed in the radiance measurements to identify the row anomaly contained pixels, referred as to the KNMI flag. Note that the KNMI flagging method remains unchanged from collection 3 to 4 (AURA-OMI-KNMI-L01B-0005-SD, 2021). The NASA flag for the latter is based on a statistical analysis of errors detected in the NASA OMTO3 L2 total column ozone. According to Schenkeveld et al. (2017) who compared the KNMI and NASA flagging results in the UV2 channel, two methods produce consistent flagging results over the full course of the OMI mission, but the NASA method is likely to be stricter and reliable. In this paper, row anomalies are filtered out when either OML1BRUG or OMUANC flags are flagged. The OMUFPMET and OMUFPSLV supply meteorological fields at OMI overpass positions, which is further detailed in Section 3.2 where the updates to meteorological inputs in OMPROFOZ are verified. In addition, OMI total column ozone product (OMTO3G) is used in deriving empirical correction spectra.



**Table 1 Input list of OMI data.**

| Product name | Processing level (spatial resolution/band [*]) | Collection number | Primary variables |
|---|---|---|---|
| OML1BIRR | L1B (UV1,UV2) | 4 | solar irradiance |
| OML1BRUG | L1B (UV1, UV2) | 4 | Earthshine radiance |
| OMCLDO2 | L2 (UV2) | 3 | cloud fraction, cloud pressure |
| OMUANC | L2 (UV2) | 4 | Row anomaly flag, snow ice flag |
| OMUFPMET | L2 (UV2) | 4 | Pressure profile, temperature profile |
| OMUFPSLV | L2 (UV2) | 4 | Surface pressure, surface skin temperature, Thermal tropopause pressure |
| OMLER | L3 (0.5° x 0.5°) | 3 | Monthly and yearly climatology of the Earth's surface Lambert Equivalent Reflectance (LER) |
| OMTO3G | L3 (0.25° x 0.25°) | 3 | Total column ozone |

[*] UV1, UV2, VIS represent bands and their corresponding spatial resolutions (except for OML1BIRR) 13 x 48
$km^2$, 13 x 24 $km^2$, and 13 x 24 $km^2$ at nadir, respectively.

**2.1 OMPROFOZ algorithm**
In our algorithm, two spectral windows are selected for 270-309 nm in the UV-1 band and 312-
330 nm in the UV-2 band and two UV-2 spatial pixels are co-added to match UV-1 spatial resolution.
To meet the computational budget in the previous data processing, OMI measurements were spatially
coadded in the flight direction, reducing the spatial resolution to $48 \times 52$ $km^2$ in the v1 product. In the
v2 data processing, PROFOZ will be released at $38 \times 26$ $km^2$, owing to the speed up of radiative transfer
calculations described in Section 3.7. The SAO ozone profile algorithm is composed of an optimal
estimation (OE) based inversion (Rodgers, 2000), radiative transfer (forward) model simulations, and
state-of-the-art calibrations (Figure 1).
**In the calibration process**, a cross-correlation technique is implemented to characterize in-orbit
slit functions and wavelength shift errors using a well calibrated, high resolution solar reference
spectrum. OMI has shown high wavelength stability (0.005-0.020 nm) over the mission lifetime (Bak
et al., 2019b; Schenkeveld et al., 2017; Sun et al., 2017) and thereby additional wavelength correction
is not carried out for each radiance and irradiance spectrum. The empirical correction so-called soft
calibration is applied for eliminating the systematic measurement biases in the wavelength range of 270
- 330 nm for ozone fitting and around 347 nm for the initial surface albedo/cloud fitting. This correction





was previously applied dependent on wavelength and cross-track position, but currently updated to
enable a correction for time-dependent degradation (Section 3.8).

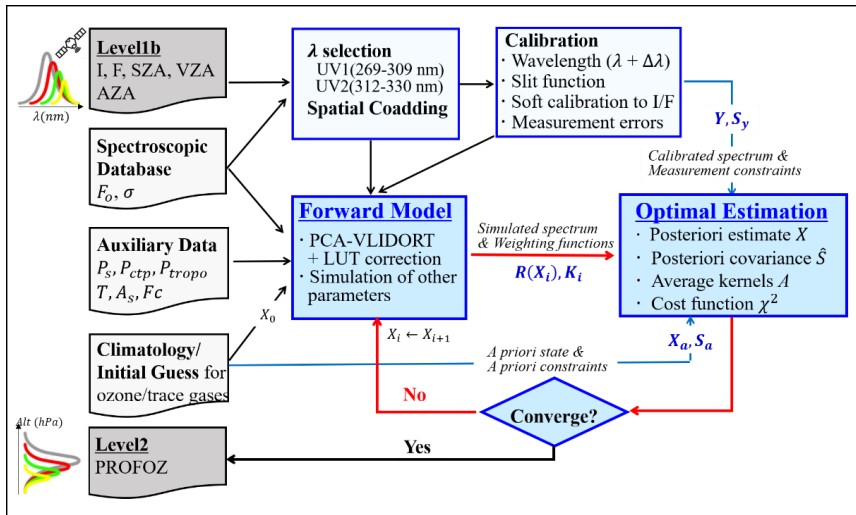

**Figure 1.** Flow chart for retrieving ozone profiles with optimal estimation-based inversion.

**This OE-based inversion** is physically regularized toward minimizing the difference between a
measured spectrum $Y$ and a spectrum that is simulated by the forward model $\mathbf{R}(X)$, constrained by
measurement error covariance matrix $\mathbf{S}_y$ and statistically regularized by an a priori state vector $X_a$
and error covariance matrix $\mathbf{S}_a$. The solution at iteration step $i + 1$ is written as
$$X_{i+1} = X_i + (\mathbf{K}_i^T \mathbf{S}_y^{-1} \mathbf{K}_i + \mathbf{S}_a^{-1})^{-1}[\mathbf{K}_i^T \mathbf{S}_y^{-1}(Y - \mathbf{R}(X_i)) - \mathbf{S}_a^{-1}(X_i - X_a)] \ , \tag{1}$$
where each component of $\mathbf{K}$ is the derivative of the forward model, called the Jacobians or weighting
function matrix. $Y$ is composed of the logarithm of the sun-normalized radiance. To construct $\mathbf{S}_y$, the
normalized random-noise errors of radiance and irradiance taken from OMI L1b products are summed
up as total measurement errors. The measurement errors are typically underestimated and then noise
floors (0.4 % below 310 nm, 0.15-0.2% above) are imposed on as a minimum value. Note that $\mathbf{S}_y$ is a
diagonal matrix, assuming that measurement errors are uncorrelated among wavelengths.
The optimal estimate is iteratively updated until convergence when the relative change in the cost
function between previous and current iterations is less than 1.0 %. The cost function $\chi^2$ is given by


$$\chi^2 = \left\| \mathbf{S}_y^{-\frac{1}{2}} \{ \mathbf{K}_i (\boldsymbol{X}_{i+1} - \boldsymbol{X}_i) - [\boldsymbol{Y} - \mathbf{R}(\boldsymbol{X}_i)] \} \right\|_2^2 + \left\| \mathbf{S}_a^{-\frac{1}{2}} (\boldsymbol{X}_{i+1} - \boldsymbol{X}_a) \right\|_2^2.$$
(2)

Maximum number of iterations is set to be 10 against the divergence. Typically, it takes 2-3 iterations
to converge, but increasing to 6-7 for thick clouds.
**The state vectors** to be fitted in OMPROFOZ v2 are listed in Table 2, together with their a priori
value and a priori error. Compared to OMPROFOZ v1, three kinds of parameters are newly added to
implement the slit function linearization and common mode correction as a pseudo absorber. A *priori*
value and error are set empirically for spectroscopic parameters, and are taken from climatological
datasets for geophysical parameters such as atmospheric ozone and surface albedo. They are assumed
to be uncorrelated between fitting parameters, except for a priori ozone error covariance matrix with
the correlation length of 6 km. Cloud fraction is initially taken from OMCLDO2 and fitted at 347 nm
together with initial surface albedo taken from OMLER.
**Table 2**. List of fitting variables, a priori values and a priori errors. A correlation length of 6 km is used
to construct the a priori covariance matrix for ozone variables. All the other variables are assumed to be
uncorrelated with each other.

| Fitting variales | # Variables | A priori | A priori error |
|---|---|---|---|
| Ozone at each layer | 24 | Climatology | Climatology |
| Surface albedo | 2 (1 for each channel) | Climatology | 0.05 |
| First-order wavelength-dependent term for surface albedo | 1 (only UV2) | 0.0 | 0.01 |
| Cloud fraction | 1 (only UV2) | Derived from 347 nm | 0.05 |
| Radiance/irradiance wavelength shifts | 2 (each channel) | 0.0 | 0.02 nm |
| Radiance/O$_3$ cross section wavelength shifts | 2 (each channel) | 0.0 | 0.02 nm |
| Ring scaling parameters | 2 (each channel) | -1.87 | 1 |
| offset parameters in radiance | 2 (each channel) | 0.0 | $1.0^{-4}$ |
| Slit width coefficient | 2 (each channel) | 0.0 | 0.1 nm |
| Shape factor coefficient | 2 (each channel) | 0.0 | 0.1 |
| Common mode scaling parameters | 2 (each channel) | 1.0 | 1.0 |




### 2.3 OMPROFOZ product

The previous version product was stored in the HDF-EOS5 format, but the NetCDF-4 format is applied to create the OMPROFOZ v2 product, similar to other collection 4 OMI data products. Also it is written using the TEMPO output libraries so that it shares common data structures and metadata definitions with TEMPO data products.

The main product parameters are partial ozone columns at 24 layers, ~ 2.5 km for each layer, from the surface to ~ 65 km in the unit of Dobson Unit (DU, 1 DU = $2.69 \times 10^{16}$ molecules.cm$^{-2}$). The 25-level vertical pressure grid is set initially at $P_i = 2^{-i/2}$ atm for $i=0, 23$ and with the top of the atmosphere set for $P_{24}$. This pressure grid is then modified: the surface pressure and the thermal tropopause pressure are used to replace the level closest to each one, and tropospheric layers are distributed equally with logarithmic pressure. Correspondingly, the random-noise error and solution error profiles are provided in terms of a square root of diagonal elements of random-noise error covariance matrix $\mathbf{S}_n$ and solution error covariance matrix $\hat{\mathbf{S}}$ that is directly estimated from the retrievals:

$$\mathbf{S}_n = \mathbf{G}\mathbf{S}_y\mathbf{G}^{\mathrm{T}}, \ \hat{\mathbf{S}} = \left(\mathbf{K}^T\mathbf{S}_y^{-1}\mathbf{K} + \mathbf{S}_a^{-1}\right)^{-1}, \text{ and } \mathbf{G} = \hat{\mathbf{S}}\mathbf{K}^T\mathbf{S}_y^{-1}, \tag{3}$$

where $\mathbf{G}$ is the matrix of contribution functions. The smoothing error covariance $\mathbf{S}_s$ can be also directly estimated, but is not provided in the output file. That is because it can be derived with the following relationship:

$$\hat{\mathbf{S}} = \mathbf{S}_s + \mathbf{S}_n. \tag{4}$$

$$\mathbf{S}_s = (\mathbf{A} - \mathbf{I})\mathbf{S}_a(\mathbf{A} - \mathbf{I})^{\mathrm{T}}, \tag{5}$$

where $\mathbf{I}$ is the unit vector and $\mathbf{A}$ is the matrix of averaging kernels:

$$\mathbf{A} = \frac{\partial X}{\partial X_T} = \left(\mathbf{K}^T\mathbf{S}_y^{-1}\mathbf{K} + \mathbf{S}_a^{-1}\right)^{-1}\mathbf{K}^T\mathbf{S}_y^{-1}\mathbf{K} = \hat{\mathbf{S}}\mathbf{K}^T\mathbf{S}_y^{-1}\mathbf{K} = \mathbf{G}\mathbf{K}. \tag{6}$$

A particular row of $\mathbf{A}$ describes how the retrieved profile in a particular layer is affected by changes in the true profile in all layers. It is a very useful variable to characterize the retrieval sensitivity and vertical resolution of the retrieved profile. The diagonal elements of A, known as Degrees of Freedom for Signal (DFS) represent the number of useful independent pieces of information available at each



layer from the measurement. To quantify the performance of the spectral fitting, the mean fitting
residuals are calculated for each fitting window (UV1, UV2), in the form of the root mean square of
spectral differences relative to the measured spectrum and the measured error as follows:

$$\text{RMS} = \sqrt{\frac{1}{N}\sum_1^N((I_m - I_s)/I_m)^2} \times 100 \ (\%), \quad \text{and RMSE} = \sqrt{\frac{1}{N}\sum_1^N((I_m - I_s)/I_e)^2}. \quad (7)$$

where $I_m$, $I_s$, and $I_e$ represent measured spectrum, simulated spectrum, and measured errors,
respectively, with $N$ the number of the wavelengths in each window. The RMS of fitting residuals
needs to be better than 0.2-0.3 % in the Huggins band (310-340 nm) for reliable retrievals of
tropospheric ozone (Munro et al., 1998). The RMSE describes both spectral fit quality and the stability
of regularization. The ideal value of RMSE is one. If RMSE $\ll 1$, either the fitting is overfitted or the
measurement errors are overestimated. On the other hand, if RMSE $\gg 1$, either the fitting is
underfitted or the measurement errors are underestimated.

## 3. Specification and verification of updated implementations
This section specifies new and improved updates made in the OMPROFOZ algorithm, listed in
Table 3. The corresponding impacts on the spectral fit and ozone retrievals are verified. Note that the
verification results of several implementations have already been presented in companion papers
indicated in the fourth column of Table 3, which is briefly described in this paper. The unpublished
implementations are specifically described in this paper.
**Table 3.** Lists of updates on algorithm implementations

| Implementations | OMPROFOZ v1 | OMPROFOZ v2 | Verification |
|---|---|---|---|
| A priori ozone climatology | Latitude dependent monthly profiles | Latitude and tropopause (daily) dependent monthly profiles | Bak et al. (2013) |
| Meteorological data | NCEP | OMUFPSLV(Joiner, 2023a) OMUFPMET(Joiner, 2023b) | This work |
| Irradiance | Climatological composite | Monthly composite | This work |
| Solar reference spectrum | Chance and Kurucz (2010) | Coddington et al. (2021) | Bak et al. (2022) |
| Slit function | Gaussian parameterization | Super Gaussian parameterization and linearization | Bak et al. (2019b) |
| Ozone cross section | BDM (Brion et al., 1993; Daumont et al., 1992; Malicet et al., 1995) | BW (Birk and Wagner, 2018) | Bak et al. (2020) |
| Radiative transfer calculation | VLIDORT only | PCA-VLIDORT | Bak et al. (2021) |
| Radiometric | CCD dependent soft calibration | -    CCD and time | This work |





| calibration | | dependent soft calibration<br>- Common mode correction | |
|---|---|---|---|


## 3 .1 A priori ozone climatology

An optimal estimation-based ozone retrieval can be significantly affected by the quality of a priori
data given insufficient measurement information. Therefore, the constraint can push the retrieval away
from the actual state of the atmosphere towards a priori information, especially near the boundary layer
or the tropopause where the vertical resolution of nadir satellite observations is inherently limited. In
the v1 algorithm, the a priori ozone information was taken from McPeters et al. (2007) (abbreviated as
LLM climatology) consisting of monthly average ozone profiles for every 10°-latitude zone based on
ozonesonde measurements in the troposphere and lower stratosphere and satellite measurements above.
The v2 algorithm implements a tropopause-based (TB) ozone profile climatology from which a zonal
monthly mean profile is vertically adjusted according to the tropopause height taken from the daily
meteorological database described in Sect. 3.2. Applying the TB climatology as OMI a priori was
thoroughly verified in Bak et al. (2013) who demonstrated improvements of OMI ozone profile
retrievals in comparison with ozonesondes as well as in representing the sharp gradients of ozone
vertical structures near the tropopause. Figure 2 compares tropospheric ozone retrievals on 01 February
2007 with a priori ozone constraints being taken from LLM and TB, respectively. The most noticeable
difference is identified in the northern region of Europe where abnormally high concentrations are
retrieved when LLM is used as a priori. This retrieval issue was also mentioned in comparing
OMPROFOZ v1.0 with other satellite products, data assimilation, and chemical transport model
calculation (Gaudel et al., 2018; Ziemke et al., 2014), showing large positive biases in tropospheric
column ozone during high-latitude winter, but it has not been explained. It is clearly seen that the
abnormal feature of the retrieved high ozone is closely correlated with the high LLM a priori (Fig. 2.c)
resulting from abnormally low tropopause pressure or high tropopause height (Fig. 2.e). LLM can
represent the typical vertical profiles whose ozonepause is located at ~ 8 km over high latitudes during
the winter. Therefore, with the presence of the abnormally high tropopause height, the lower
stratospheric layers of LLM profiles can be misrepresented as a priori in the upper tropospheric ozone
layers, which likely causes the large positive biases of ozone retrievals in the troposphere seen in
OMPROFOZ v1. However, an ozone profile taken from the TB climatology is re-distributed according
to the daily tropopause which becomes an ozonepause of TB profiles. In the subtropical region, LLM
may also provide incorrect information in the presence of high tropopause height, but ozone retrievals
are less affected, implying that OMI retrievals are less constrained by the a priori information in this





case due to more measurement information, unlike in the northern high-latitudes.


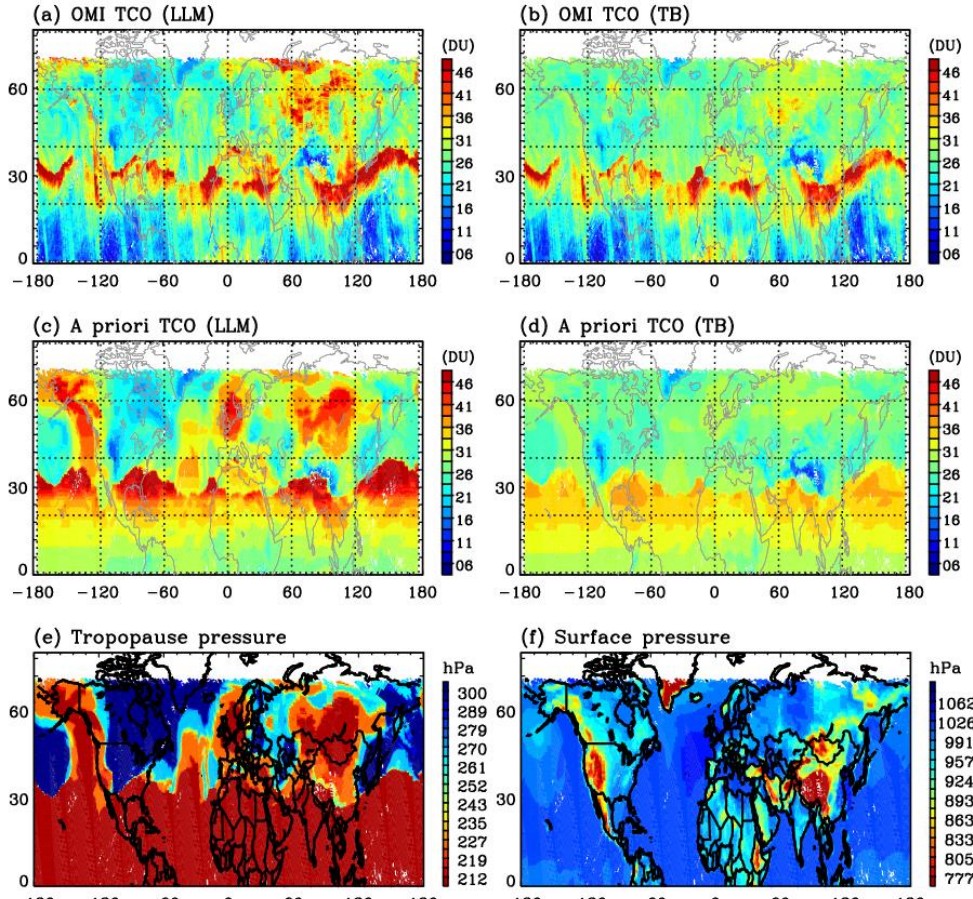


**Figure 2**. Comparison of (a,b) OMI tropospheric column ozone (TCO) and (c,d) the corresponding a priori TCO
taken from monthly and zonal mean climatologies (LLM/left, TB/right), respectively, in the Northern hemisphere
on 01 February 2007. (e) tropopause and (f) surface pressure fields are presented in the bottom panels.

**3.2 Meteorological data**

As a forward model input, the surface pressure is required to define the bottom of the atmosphere,



with the air temperature profile to account for the temperature dependence of the ozone absorption cross
section, especially in the Huggins band. The tropopause pressure is also required to be used as one of
the retrieval vertical levels to separate stratospheric ozone from tropospheric ozone, and determine the
a priori ozone profile in the case of using the TB climatology in v2. These meteorological variables
were taken externally from National Centers for Environmental Prediction (NCEP) reanalysis data
(http://www.cdc.noaa.gov), which provide 6-hourly (4 time a day) global analyses at 2.5 ° x 2 ° grids
with 17 vertical pressure levels below 10 hPa. These databases were pre-interpolated to 1:45 PM local
solar time when OMI is crossing at equator and OMI's ground pixels using nearest neighbor
interpolation and then manually transmitted to OMI SIPS. However, the data transmission has been
accidently halted since June 2011 and hence climatological monthly mean data have been used as a
back-up in the data processing. To avoid this risk, the meteorological input is switched to the internal
meteorological products, geo-collocated to OMI UV-2 1-Orbit L2 Swath from the 2D Time-Averaged
Single-Level Diagnostics (OMUFPSLV) (Joiner, 2023a) and the GEOS-5 FP-IT 3D Time-Averaged
Model-layer Assimilated data (OMUFPMET) (Joiner, 2023b). We take the average air temperature
given at 72 pressure levels above the center of the ground pixel from OMUFPMET as well as surface
temperature, surface pressure, and thermal tropopause pressure at the center of the ground pixel from
OMUFPSLV. The impact of switching meteorological input on the spectral fitting residuals is
insignificant (not shown here), implying that the residuals might be absorbed by other state vectors.
Figure 3 illustrates that ozone profile retrievals are changed by 2-3 DU, especially in the tropopause
region due to changes of a priori ozone profiles in adjusting the climatological TB ozone profile around
the daily tropopause height.

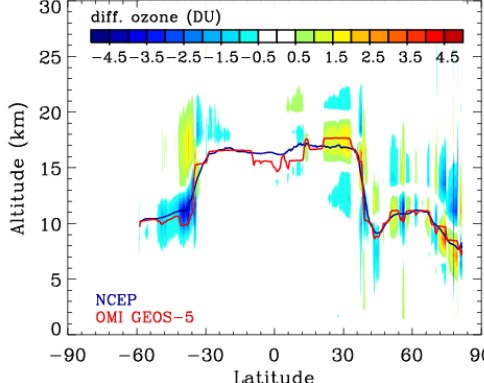


**Figure 3.** Differences of OMI ozone profile retrievals (DU) along the nadir view from 7[th] orbit of measurements
on 15 Jun 2006, due to switching the meteorological input from NCEP to OMI GEOS-5 (OMUFPSLV and
OMUFPMET). The solid line represents the tropopause height from NCEP (blue) and OMI GEOS-5 (red).

### 3.3 Ozone cross section

The BDM cross-section measurements have been the standard input for retrieving ozone profiles using BUV measurements over the last decade (Liu et al., 2013, 2007; Orphal et al., 2016). In a companion paper (Bak et al., 2020), the new BW ozone cross-section dataset was tested to check if there is room to improve our ozone profile retrievals, which made us switch the cross section from BDM to BW in OMPROFOZ v2. As illustrated in Figure 4 (upper), the BW dataset provides improved temperature coverage from 193 K to 293 K, every 20 K over the BDM dataset given only at five temperatures above 218 K. Therefore, BW measurements were better parameterized as quadratic temperature-dependent coefficients with uncertainties of 0.25-2 % whereas for BDM measurements fitting residuals of 2-20 % remains. Note that parameterized coefficients of cross-section measurements are typically applied in both column ozone and ozone profile retrievals for conveniently representing the temperature dependence of cross-section spectrum. Bak et al. (2020) also demonstrated the improved performance of ozone profile retrievals through comparison with ozonesonde measurements, showing a significant reduction of the standard deviations, by up to 15 % in the lower stratosphere and upper troposphere where atmospheric temperatures are lower than ~ 200 K.

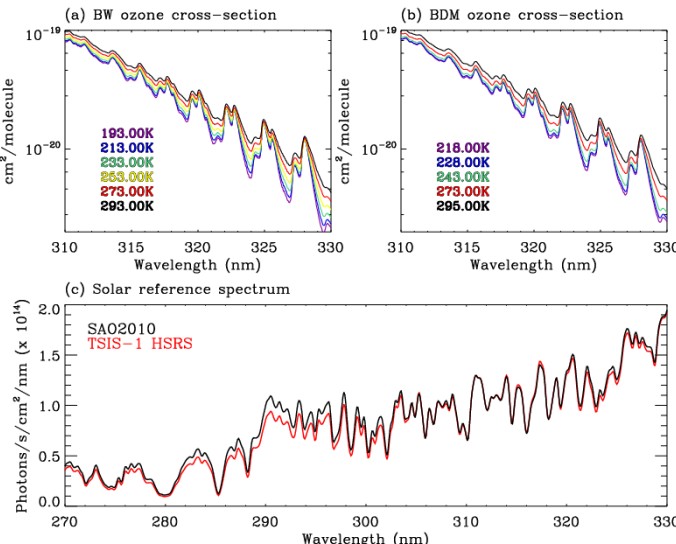

**Figure 4.** Comparisons of (a.b) ozone cross-sections and (c) solar reference spectrum used in OMPROFOZ v1 and v2 algorithms. Note that high-resolution solar reference spectrum is convolved with a Gaussian slit function of 0.4 nm FWHM (Full Width at Half Maximum) resolution.



### 3.4 High-resolution solar reference spectrum


An accurate, high-resolution extraterrestrial solar reference spectrum is required for either
wavelength calibration or slit function characterization. We decided to switch the solar reference
spectrum from Chance and Kurucz, (2010) to Coddington et al., (2021). Figure 4.c illustrates
radiometric discrepancies between the new solar reference called the TSIS-1 Hybrid Solar Reference
Spectrum (HSRS) and the old solar reference called the SAO2010. A companion paper evaluated that
the radiometric uncertainties of the new reference spectrum are below ~ 1 % whereas for SAO2010
those range from 5% in the longer UV part to 15 % in the shorter UV part (Bak et al., 2022). Furthermore,
they confirmed an opportunity to improve the spectral fitting of slit functions and hence the spectral
fitting of ozone when using the TSIS-1 spectrum; the impact on ozone profile retrievals is 5-7 % in the
troposphere.

### 3.5 Solar irradiance spectrum


OMI makes solar irradiance measurements near the northern hemisphere terminator of an orbit once
per day, which are required to calculate top-of-atmosphere reflectance and to estimate an on-orbit slit
function in ozone profile retrievals. In order to reduce the short-term noise of OMI individual
measurements, the v1.0 algorithm implemented the use of climatological solar spectra derived from
three years of daily OMI Level 1B product (2005-2007). In the v2.0 algorithm, irradiance spectra are
tabled as a monthly average to reduce the short-term noise as well as cancel out the common degradation
existing in radiance and irradiance. Figures 5 and 6 compare irradiance measurements averaged over
July for each year from collection 4 and collection 3, respectively. Collection 3 shows significant short-
term noise in daily measurements in the UV2 range, around 3-5 % and also systematically decreasing
patterns of monthly irradiance spectra from – 10 % in the UV1 range and -6 % in the UV2 range over
the mission. Collection 4 provides much improved irradiance spectra with respect to both degradation
and noise errors. In addition, OMI random-noise errors in the monthly average spectra are compared.
Collection 4 ranges from 0.02 % in the UV1 and 0.04 % in the UV2, consistently over the mission.
However, collection 3 shows somewhat different features in the UV2 range, like more wavelength
dependence and a systematic drift as of 2008-2009. Figure 7 shows the impact of switching OMI level1b
product from collection 3 to 4 on fitting residuals resulting from ozone profile retrievals on 16 July
2020; the average fitting residuals are plotted as a histogram for each fitting window. In this experiment
the v2 implementations are identically applied without radiometric corrections (soft calibration and
common mode correction are turned off). In addition, the impact of using monthly and daily irradiance
is investigated. As shown, fitting residuals are noticeably improved in both fitting windows due to





switching from collection 3 to 4. This experiment illustrates that monthly irradiances should be used

instead of daily measurements when using the collection 3 product. In comparison, the corresponding

impact on fitting residuals with collection 4 product is not very significant due to improvements of

short-term noise errors in daily irradiance measurements, but the number of retrievals with smaller

fitting residuals increases in the UV2 band.

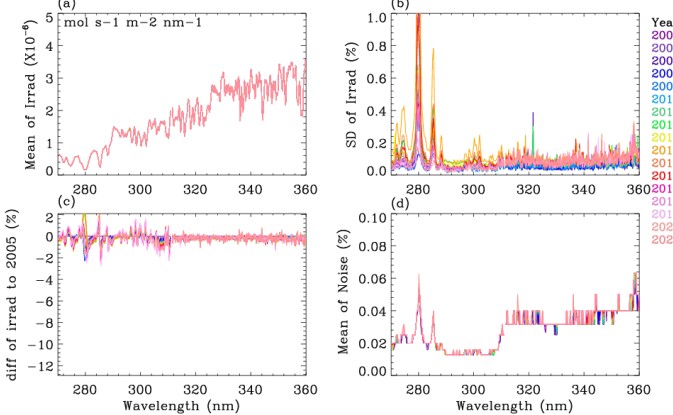

**Figure 5.** (a) Monthly mean irradiance spectra of OMI collection 4 product in July from 2005 to 2021 at the 10$^{th}$ cross-track position for UV-1 band and 20$^{th}$ cross-track position for UV-2 band without coadding. (b) Corresponding standard deviations of the monthly mean irradiances, (c) Biases of the mean irradiances relative to 2005, and (d) Monthly mean random noise errors.

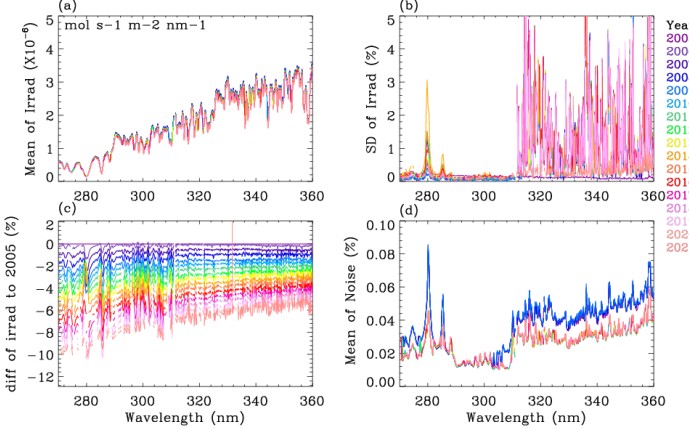

**Figure 6**. Same as Figure 5, but for OMI collection 3 irradiance product.

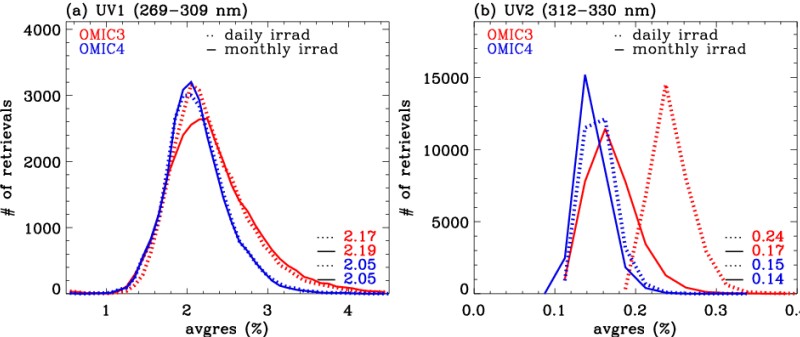

**Figure 7**. Histograms of average fitting residuals from OMI collection 3 (red) and 4 (blue) level 1b products on 15 July 2020, in (a) UV1 and (b) UV2 ranges, respectively. In order to make a fair comparison, this experiment limits OMI measurements to the western side of the swath to avoid using row anomaly cross-track pixels and empirical recalibration is not applied. Fitting residuals are evaluated with both daily (dashed) and monthly mean (solid) OMI irradiance measurements. The median values of average fitting residuals are presented in the legend.

### 3.6 Instrument spectral response function (ISRF) parameterization and linearization

OMI ISRFs were previously parameterized as a standard Gaussian by fitting the slit width ($w$) from OMI solar irradiances separately for each channel and each cross-track position. In the updated implementation, one more parameter, shape factor ($k$) is added to parameterize ISRFs as a Super Gaussian ($S(\Delta\lambda) = exp\left[-\left|\frac{\Delta\lambda}{w}\right|^{k}\right]$). However, slit functions in radiance could deviate from those derived from solar spectra due to the sensitivity to scene heterogeneity, differences in stray light between radiance and irradiance, and intra-orbit instrumental changes. These might cause some spectral structures in the radiance fitting. Therefore, the v2 algorithm treats these spectral errors as Pseudo Absorbers (PAs), which is derived as $\frac{\partial I}{\partial p} = \frac{\partial S}{\partial p} \otimes I_h$ (p = $w$ or $k$) through the slit function linearization. As specified in Table 2, these PAs are iteratively adjusted with zero-order scaling parameter. The description and evaluation of this implementation for OMI ozone profile retrievals is detailed in a companion paper (Bak et al., 2019b).

### 3.7 Radiative Transfer Calculation

The radiative transfer (RT) model is needed for calculating the forward model component such as top-of-the-atmosphere radiances, and Jacobians of radiances with respect to the atmospheric and surface parameters. The radiance calculation is made for a Rayleigh atmosphere (no aerosols) with Lambertian





reflectance assumed for the surface and for clouds. The Independent Pixel Approximation (IPA) is
employed to treat partial clouds by assuming a cloud reflectivity of 80 %: $I = I\ (R_{sfc}, P_{sfc})(1 - f_c) +$
$I\ (R_{cloud}, P_{cloud})$. Individual radiances need to be simulated at finer grids than at least 4 pixels per
FWHM so that the spectral convolution is applied to account for OMI spectral resolution. To reduce the
computational burden, a few wavelengths are effectively selected $(\lambda_e)$ for running RT model and then
interpolated to regular high-resolution grids $(\lambda_h)$ with the radiance adjustment for errors caused by the
spectral resolutions as follows:
$$I(\lambda_h) = I(\lambda_e) + \sum_{l=1}^{N} \frac{\partial I\ (\lambda_e)}{\partial \Delta_l^{gas}}\left(\Delta_l^{gas}(\lambda_h) - \Delta_l^{gas}(\lambda_e)\right) + \frac{\partial I\ (\lambda_e)}{\partial \Delta_l^{ray}}\left(\Delta_l^{ray}(\lambda_h) - \Delta_l^{ray}(\lambda_e)\right), \qquad (7)$$
where $\frac{\partial I}{\partial \Delta_l}$ represents for Jacobians with respect to optical properties at layers $l$ ($l = 1$ to N). In the v2
forward model, both $\lambda_c$ and $\lambda_h$ are set to be finer than intervals previously used as noted in Table 4
where the implementation details between v1 and v2 forward models are compared. To accelerate RT
calculations, the RT model is switched from VLIDORT v2.4 to PCA-based VLIDORT v2.8; multiple
scattering (MS) calculations are performed at individual wavelengths in the former whereas MS
calculations in the latter are performed only for a few EOF-derived optical states which are developed
from spectrally binned sets of inherent optical properties that possess some redundancy. In both v1 and
v2 forward models, the polarization is not part of the direct RT simulation of the entire spectrum, but a
polarization correction is applied to speed up the RT. In the v1 forward model, vector calculations are
additionally executed at 14 wavelengths to calculate the scalar vs. vector differences at these
wavelengths which are then interpolated to every wavelength. However, residual polarization errors
remain along with other approximation errors arising from using 4 half-streams and coarse vertical
layers (~ 2.5 km thick). The v2 forward model reduces the number of half streams from 4 to 2 with a
resulting increase in speed of a factor of ~2. To eliminate the increasing RT approximation errors, a
look-up table (LUT)-based correction is performed, which enables to adjust the differences in RT
variables due to different number of streams (2 vs. 6) and number of layers (24 vs. 72) as well as
neglecting polarization effect. As verified in a companion paper, updates improve the retrieval speed by
a factor of ~ 3.3 as well as the retrieval accuracy (Bak et al., 2021). Note that the Ring simulation
remains unchanged from v1 algorithm; the spectral structure of the Ring signal is externally simulated
with the iterative fitting of amplitude of the Ring spectrum and then subtracted from the measured
spectral reflectance.




### 3.8 Soft calibration


The left panels of Figure 8 show (a) the spectral fitting residuals averaged in the latitude band of 60ºS
to 60ºN, (b) tropospheric column ozone (TCO) distribution, and (c) cross-track dependent stripe errors
of TCOs where OMI collection 4 L1b product is applied without any radiometric corrections. As shown,
there remain quite persistent residuals of up to ~ 1.0 % in the UV1 range and of up to 0.3 % in the UV2
range. The TCO distribution shows the along-track stripes that are commonly found in OMI trace gas
products. The cross-track dependent stripes of TCO are evaluated for 18 bands of latitude, as anomalies
in the ratio of each cross-track column to the average column taken within cross-track positions 5-25
(1-based). The amplitude of anomalies is within $\pm 10$ % at nadir pixels, but reaching to 40 % at off-
nadir pixels, with some dependency on latitudes. However, stratospheric column ozone (SCO) retrievals
are almost free of stripe errors (not shown here). To reduce the striping, a soft calibration was applied
to OMI radiances in OMPROFOZ v1. The soft spectra are derived as a systematic component of
differences between measured and simulated radiances at tropical clear-sky pixels in summer where the
forward model calculations are more accurate to attribute the residuals to measurement biases. The soft
spectra are re-derived for OMI collection 4 L1b product using the v2 forward model calculations (Sect
3.7). The ozone profile input is prepared from 10-degree zonal averages of daily MLS measurements
above 215 hPa and climatological ozone profiles taken from McPeters and Labow (2012) below. The
climatological profile shape is adjusted to account for the daily variability using 10-degree zonal
averages of the level 3 OMI TOMS-like total ozone product (OMTO3d). To smooth out the impact of
daily ozone variabilities, one-week measurements during July 11-17th over the tropics 20°S-20°N are
used in deriving the soft spectra after screening out outliers of extreme viewing geometries (SZA > 60°),
cloudy pixels ($f_c$ < 0.2), bright surfaces ($A_{sfc}$ > 0.1), and aerosol contaminated pixels (aerosol index
> 5) as well as abnormally large values of average residuals (UV1 > 8, UV2 > 3). Note that the threshold
value of filtering out aerosol pixels needs to be relaxed due to the overestimation errors of aerosol index
at initial iteration. Figure 9 displays the cross-track dependent soft spectrum for the case of July 2005
when instrument degradation is negligible and row-anomaly damage has not occurred. It illustrates the
existence of systematic residuals between measured and simulated radiances within 2 % in UV2 and
mostly from -7 to 3 % in the UV1, except for some spikes. The right panels of Figure 8 demonstrate
how soft calibration works for improving ozone retrievals in comparison to the left panels where soft
calibration is tuned off. It is clearly shown that the systematic spikes are mostly eliminated as well as
cross-track dependent stipes are globally reduced even up to high-latitudes. In particular, the "anomalies"
are reduced to within 0.1 %, except at first cross-track pixels. This calibration has been applied
independent of time and latitude in the v1 algorithm. To account for OMI instrument degradation errors,

the v2 soft spectra are developed for every year. As an example, the yearly soft spectra are displayed at
several cross-track positions in Figure 10. There is noticeable yearly variation in the UV1 band,
typically within 2-3% over 17 years. The most significant degradation features are found at the first
cross-track pixel in the UV1 band, with relative change of 5 % or more. For cross-track positions 13,
18, 22, correction spectra cannot be derived for most of the time periods after 2008 due to the occurrence
of serious row anomaly. Although correction can be derived for cross-track position 13 during 2020, it
is significantly different from those before 2008, indicating that it is still affected by row anomaly. The
yearly variation in the UV2 band is much smaller, and can be clearly identified below ~315 nm to be
within 1 %. However, it could make a significant impact on ozone profile retrievals because the spectral
fit residuals need to be smaller than 0.2-0.3 % in the Huggins band for reliable retrieval quality of the
tropospheric ozone (Munro et al., 1998).

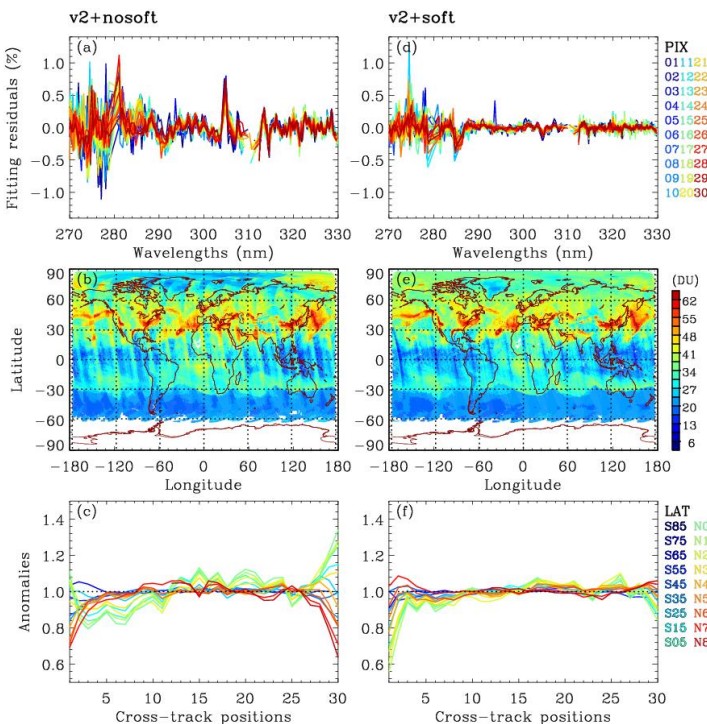

**Figure 8**. (a, d) Spectral fitting residuals (%) averaged in the latitude of 60ºS and 60ºN from OMI measurements
on 15 June 2006, (b,e) the global distribution of tropospheric column ozone (TCO, DU), and (c,f) anomalies of
TCO as a function of 18 latitude bands. Left and right panels are for without and with soft calibration, respectively.

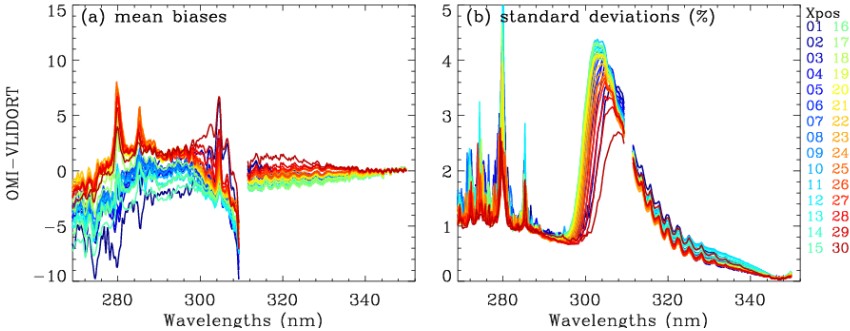

**Figure 9** (a) soft calibration spectra derived for collection4 OMI L1b products in July 11-17, 2005, representing the systematic biases between measured and simulated spectrum. (b) the standard deviations of the systematic biases, representing the uncertainties of soft calibration spectra.

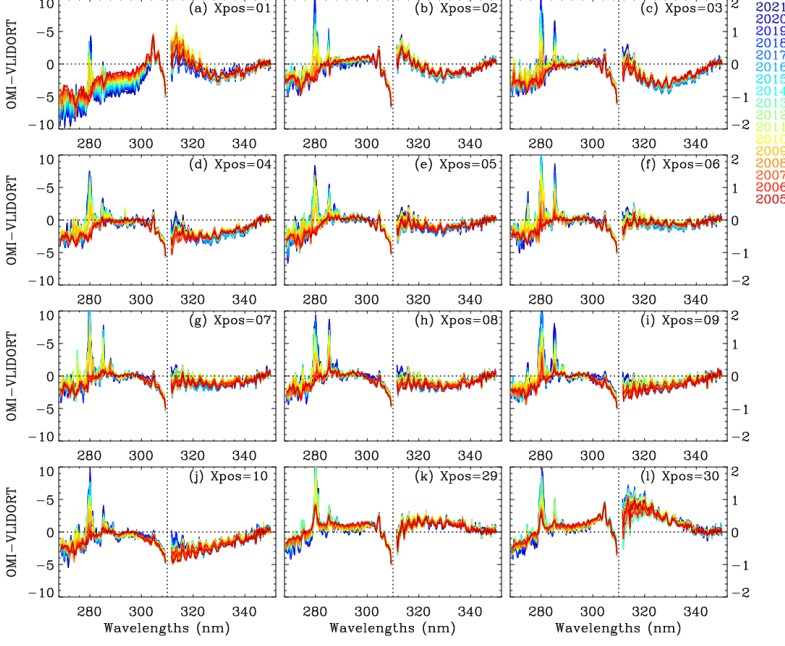

**Figure 10.** Yearly dependent soft calibration spectra from 2005 to 2021 at several cross-track positions (Xpos, UV1-based) which have been not affected by row anomalies over the mission. Note that the UV1 and UV2 bands are plotted with different Y-axis ranges (left Y-axis for UV1 and right Y-axis for UV2) for better visualization.



### 3.9 Common mode correction

As compared in Figures 11 left and middle panels, the soft calibration is less effective in eliminating the systematic residuals at high solar zenith angles, especially in the UV2 band where the spectral residuals vary from 0.1 % at lower SZAs to 0.4 % at higher SZAs. This implies the existence of a spectral dependence of the radiometric calibration and detector sensitivity on the signal represented by solar zenith angle, which is not accounted for in the soft calibration dependent only on CCD dimension. Moreover, the soft calibration induces the systematic errors spiking at around ~ 285 nm and 305 nm in the UV1 band. A common mode correction (CMC) is newly implemented in OMPROFOZ v2, to correct the remaining radiometric errors. The common mode spectrum of the fitting residuals is physically treated as a pseudo absorber, along with a scaling coefficient that is iteratively fitted in each of the UV1 and UV2 windows. Therefore, the scene-dependent radiometric errors could be partly accounted for. This kind of correction is originally used in the spectral fitting process where a common mode residual could be calculated on-line for each orbit of measurement. However, additional on-line calculation is not practical for the time-consuming optimal estimation-based ozone profile retrieval process. Therefore, we derive time-independent common mode spectra by averaging three days of fitting residuals (July 13$^{th}$ -15$^{th}$, 2005) over five solar zenith angle regimes [0°-40°, 40°-60°,60°-70°, 70°-80°, 80°-85°] for each cross-track position. As demonstrated in Figure 11 right panel, the applied common mode spectrum is likely to absorb the remaining spectral errors and hence the fitting accuracy is globally improved. For example, the systematic features are clearly reduced above 285 nm in the UV1 window, but the noisy features are still not well fitted below 285 nm. In the UV2 band, applying CMC reduces the dependence of fitting residuals on both solar zenith angle and cross-track pixels and hence the remaining residuals are globally less than 0.1 % at most wavelengths. As shown in Figures 12, striping patterns of tropospheric ozone retrievals could be reduced due to improvements of retrievals at the first cross-track pixels in the tropics where soft calibration deepens anomalies (Figure 8.f). Comparisons with OMPROFOZ v1 retrievals (Figure 12.d-f) demonstrate that OMPROFOZ v2 product provides global information on tropospheric column ozone with smaller retrievals biases due to radiometric calibration errors and more consistent data quality with respect to different viewing geometries and latitude.

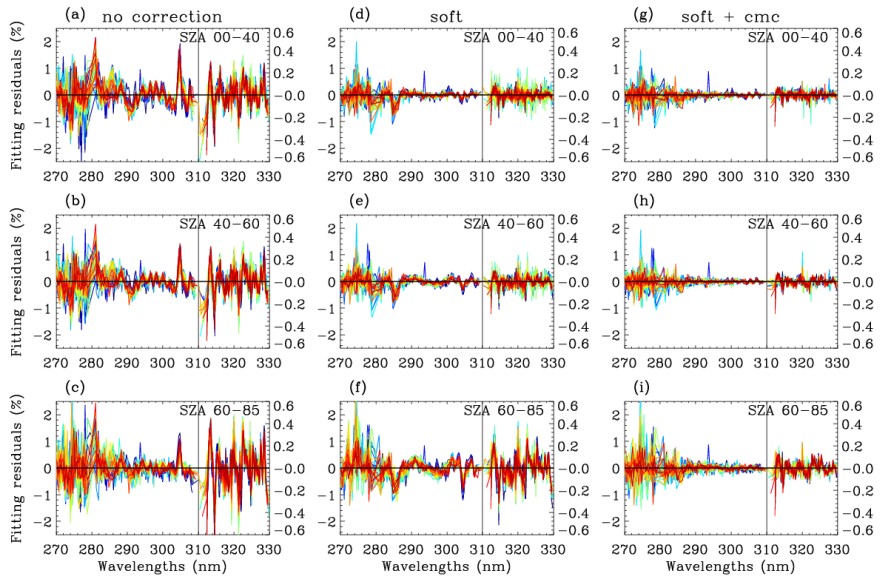

527

**Figure 11**. Comparison of spectral fitting residuals (%) averaged for three solar zenith angle regimes (00º-40º, 40º-60º, 60º-85º) from OMI measurements on 15 Jun 2005, with different radiometric calibration settings (left: all radiometric correction is turned off, middle: soft calibration is turned on, right: soft calibration and common model correction are turned on). Note that the residuals are plotted in different y-axis range below (left y-axis) and above (right y-axis) 310 nm, respectively.

533

## 4. Validation with ozonesonde measurements

535
Comparisons against ozonesonde measurements are performed to highlight improvements of data quality and long-term consistency of OMPROFOZ v2 over OMPROFOZ v1. Ozonesonde measurements are obtained from three sites over central Europe during the period of 2005 to 2020, listed in Table 5. Balloon-borne ozone profiles are regularly measured two/three times per week at these sites located close to each other. The coincidence criteria used to pair OMI and ozonesonde measurements are within 100 km and 6 hours and then the closest pair is selected after screening out row anomaly flagged pairs. For comparison, individual ozonesonde soundings are converted from mPa into DU and then interpolated at OMI vertical grids, but without adjusting the vertical resolution into OMI to address the total errors of OMI retrievals including smoothing errors. The relative difference is calculated as (OMI-sonde)/ sonde $\times$ 100 %. Extreme values that are beyond the mean by $3\sigma$ are dropped in estimating the comparison statistics.



547  Figure 13 shows comparisons of ozone profiles between OMI and ozonesonde during the pre and

548 post Row Anomaly (RA) periods, respectively. The pre-RA period is set to be from the beginning of the

549 mission through 2008 when the row anomalies were relatively not serious and the post-RA period is

550 after that. Both v1 and v2 retrievals are positively biased relative to ozonesonde measurements. The

551 mean biases (MBs) of profile differences are less than 20 % over the layers when OMPROFOZ v2

552 profiles are compared during the pre-RA period. On the other hand, MBs of OMPROFOZ v1 are largely

553 skewed by ~ 45 % in the tropopause region. The comparison also confirms significant improvements

554 of OMPROFOZ v2 retrievals, with the reduction of standard deviations (SDs) by ~ 40 % around the

555 tropopause. These improvements are achieved mainly due to implementing TB ozone profile

556 climatology which could better represent the profile shape in the UTLS as mentioned in Section 3.1.

557 Comparison statistics between OMPROFOZ v2 and ozonesondes profiles are generally consistent

558 before and after the RA occurrence in spite of the inconsistent sampling resulting from the occurrence

559 of RA so that only about half of the OMI measurements remain valid, mostly on the west of nadir during

560 the post-RA period. However, OMPROFOZ v1 profiles are shown to be much more affected by

561 temporal changes of OMI instrumental stability, especially in the lower atmosphere.

562  The rest of this section is concentrated on assessing the consistency of tropospheric ozone retrieval

563 quality with respect to temporal changes. For this comparison, tropospheric ozone columns (TCOs) are

564 integrated over the troposphere between 200 hPa and 900 hPa from ozone profiles to avoid the impact

565 of different meteorological inputs used in V1 and V2 retrievals. In order to check the seasonal changes

566 of retrieval quality, comparison statistics of tropospheric ozone between OMI and ozonesondes are

567 derived for each month during the pre-RA period. The seasonal changes of retrieval quality could be

568 mainly related to the solar zenith angle dependency of OMI measurement sensitivity to the lower

569 tropospheric ozone, which also causes the inconsistency of retrieval quality between lower and higher

570 latitudes. As shown in Figure 14.a, monthly biases of OMI TCO are minimized below ~ 2 DU from

571 June to October when the solar zenith angles are relatively small, commonly for OMPROFOZ v1 and

572 v2. However, the mean biases of OMPROFOZ v1 increase up to ~ 6-9 DU during January-March, while

573 OMPROFOZ v2 show the moderate change of monthly biases from winter to summer, with the smaller

574 SDs of TCO differences by ~3-4 DU during December-March (Fig. 14.b).

575  In order to check the long-term stability, TCO differences are averaged into four seasons for each

576 year from 2005 to 2020 in Figures 14.c and d. The existence of a long-term drift is clear with MBs of

577 OMPROFOZ v1 TCO decreasing from ~ 4.35DU before 2008 to ~ 0.05 DU after 2015. This temporal

578 drift is largely corrected in OMPROFOZ v2 retrievals and the standard deviations of TCO differences

579 are reduced generally over the entire period. In addition, OMPROFOZ v1 shows more spikes in both





MBs and SDs than OMPROFOZ v2, especially during the period of 2011 to 2015 when the RA
dynamically expands. Those spikes could be attributed to row anomaly-contaminated retrievals
unscreened with the KNMI-based row anomaly flags (used in v1) which is considered to be less strict
than TOMS-based row anomaly flags (used in v2).

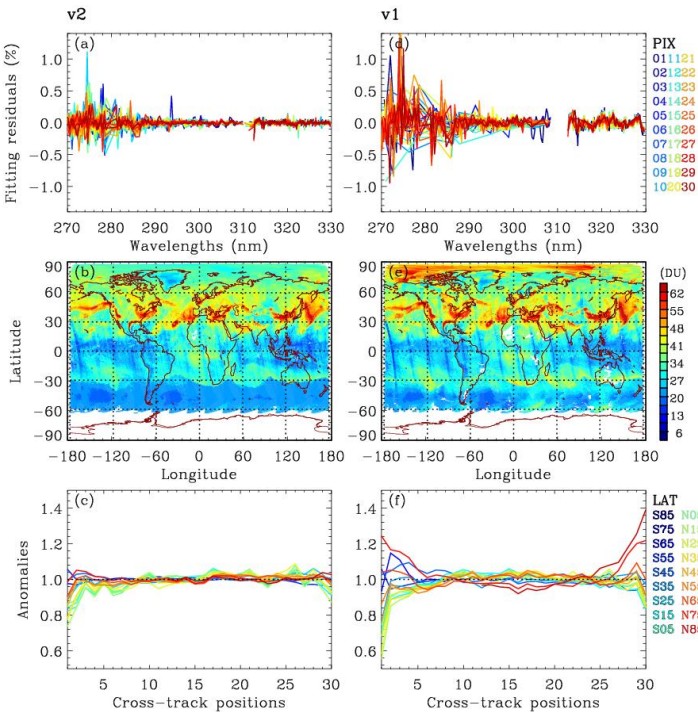


**Figure 12**. Same as Figure 8, but for V2 (OMI collection 4 product with the final v2 algorithm) and V1 (OMI
collection 3 with the v1 algorithm).







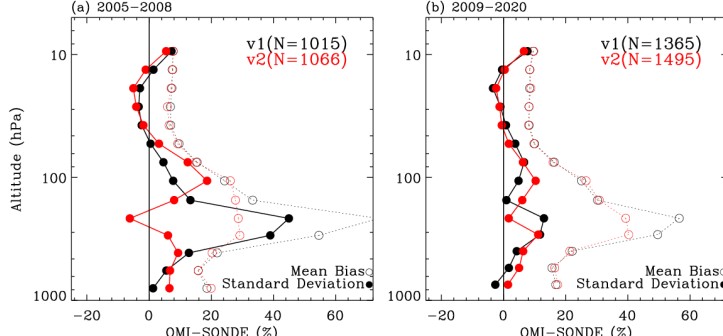


**Figure 13**. Comparisons of ozone profiles between OMI and ozonesonde during (a) pre-row anomaly and (b) post-row anomaly periods, respectively. OMI retrievals are qualified with RMSE < 3, RMS < 2%, and cloud fraction less than 0.6. The number of coincident pairs (N) is given in legend.


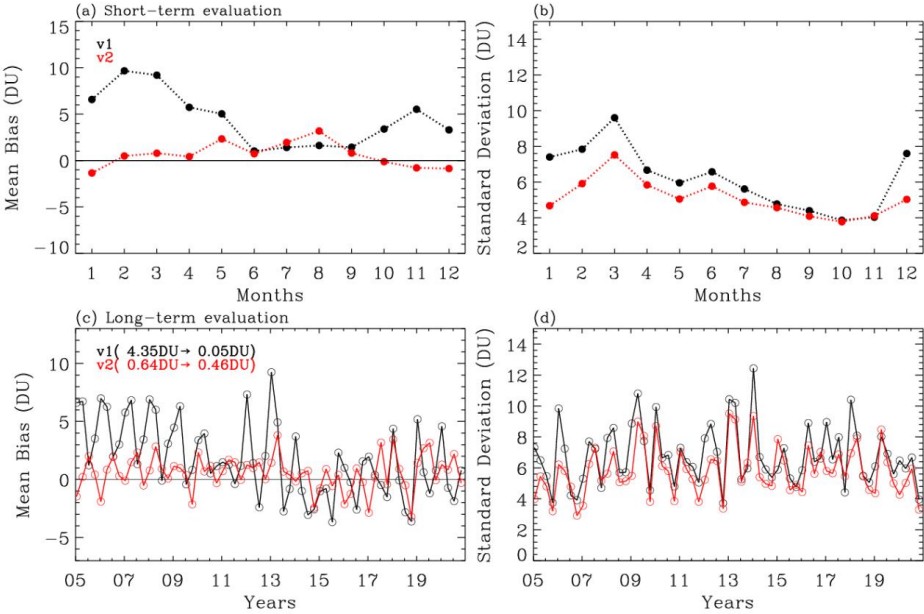


**Figure 14**. (a) Monthly mean and (b) corresponding standard deviations in differences of tropospheric column ozone (TCO, 200-900 hPa) between OMI and ozonesondes during the period of 2005 to 2008. (c,d) is same as (a,b), but for seasonal differences of TCO from 2005 to 2020. The legend of Fig. c represents the overall mean for the period of 2005-2008 and 2015-2020, respectively.



## 5. Summary and Conclusion

The Smithsonian Astrophysical Observatory (SAO) ozone profile retrieval algorithm has been run in NASA's Science Investigator-led Processing System (SIPS) to create the Ozone Monitoring Instrument (OMI) ozone profile (OMPROFOZ) research product. Since the first data release, the efforts to improve the retrieval accuracy and long-term consistency of OMI ozone profile retrievals have continued externally. In this paper, the second version of OMPROFOZ research product is introduced, which will be released at GES-DISC while the first version will continue to be archived at AVDC. One of the major changes is to switch the L1b data from collection 3 to collection 4, for both radiance and irradiance as well as the accompanying auxiliary datasets. We also changed several geophysical and spectroscopic inputs including meteorological data, ozone profile climatology, high-resolution solar reference spectrum, and ozone absorption cross-section dataset. Implementations of forward model calculations and measurements calibrations are improved. The v2 forward model employs a faster principal component analysis (PCA)-based VLIDORT model, along with the LUT-based correction which speeds up the online radiative transfer model calculation while corrections to the approximation produce improved accuracy. The resulting speed-up allows OMI native measurements to be processed for OMPROFOZ v2, with data resolution of $48 \times 26$ km$^2$ at nadir. Note that to meet the computational cost, the previous data were processed after coadding OMI measurements at the spatial resolution of $48 \times 52$ km$^2$. To better represent the shape of OMI slit functions, the slit width and shape factor are parameterized from OMI irradiances, assuming a super Gaussian, instead of a normal Gaussian. Moreover, the effects of slit function differences between radiance and irradiance on ozone retrievals are accounted for as pseudo absorbers in the iterative fit process. The OMI irradiance measurements are included via a monthly average instead of a 3-year climatological mean to cancel out the degradation offset between radiance and irradiance measurements. The empirical soft calibration spectra are re-derived annually to be consistent with the updated implementations to remove the systematic differences between measured and simulated radiances. "Common mode" correction spectra are derived from remaining residual spectra after soft calibration with the dependency on solar zenith angle. The common mode is included as a pseudo absorber in the iterative fit process, which helps to smooth out the discrepancies of ozone retrieval accuracy between lower and higher solar zenith angles and between nadir and off-nadir pixels.

In order to verify improvements of OMPROFOZ data quality, both v1 and v2 ozone profiles are evaluated against ozonesonde measurements taken from central Europe. It is clearly shown that ozone profile retrievals are greatly improved in the troposphere, especially around the tropopause, with the reduction of mean biases by ~ 25 % during the pre-RA season. The standard deviations of mean biases





are also improved by ~ 40 % and ~ 20 % before and after the RA occurrence. The comparison with
ozonesondes also confirms that the temporal consistency of tropospheric ozone quality is improved.
The seasonal change of data quality from summer to winter is predominant in OMI tropospheric ozone
with V1 processing. However, OMPROFOZ v2 data quality shows much better consistency, with the
seasonal changes of retrieval biases within ~ 2-3 DU. Above all, we validate that the OMI long-term
degradation is better accounted for in OMPROROZ v2 processing, along with switching OMI L1b data
from collection 3 to collection 4 and updating implementation details. In OMPROFOZ v1, mean biases
of tropospheric ozone relative to ozonesonde shows a drift in errors from 4.35 DU to 0.05 DU before
and after the RA occurrence, which are greatly reduced to within ± 0.5 DU for both periods in
OMPROFOZ v2.
This new algorithm has been delivered to the NASA OMI SIPS for operational processing and the
reprocessing of the entire mission is in progress. The OMPROFOZ v2 product will be distributed via
the NASA GES DISC.

**Author Contributions** J.B and X.L designed the research. X.L developed the OMPROFOZ v1
and J.B updated it to OMPROFOZ v2. K.Y contributed to improving the forward model simulations
and transferring codes into SIPS; G.G.A and E.O.S developed the reading modules for OMI collection
4 products; K.C advised the update to solar reference spectrum; C.H.K provided financial support to
make this study continue. J.B and X.L conducted the research and wrote the paper; all authors
contributed to the analysis and writing.

**Competing interests**. The authors have no competing interests

**Acknowledgement**
Both calculations and simulations are done on the Smithsonian Institution High-Performance Cluster
(SI/HPC) (https://doi.org/10.25572/SIHPC). We acknowledge the WOUDC for providing ozonesonde
data, OMI science team for providing OMI collection 3 and OMI collection 4 products. We would like
to thank David Haffner and Zachary Fasnacht for providing useful comments regarding OMI collection
4 products.

**Data Availability**
All OMI datasets are available at https://disc.gsfc.nasa.gov/ (last access: 15 July 2023). The ozonesonde
data used to validate our ozone profile retrievals were obtained though the WOUDC. The WOUDC
dataset is available at https://woudc.org/ data/products/ozonesonde/ (last access: 15 July 2023).

**Financial support.** This research has been supported by NASA Aura science team program (grant
no. NNX17AI82G and 80NSSC21K0177) and Basic Science Research Program through the National
Research Foundation of Korea(NRF) funded by the Ministry of Education (grant no.
2020R1A6A1A03044834 and 2021R1A2C1004984).

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
