# Peer review of "An improved OMI ozone profile research product version 2.0 with collection 4 L1b data and algorithm updates"

_Atmospheric Measurement Techniques, 2023_

## Author Comment (AC1)

**Overall Comments:**

The second version of the OMPROFOZ research product (OMPROFOZ v2) has been introduced in this paper, which incorporates several improvements to enhance the accuracy and long-term consistency of ozone profile retrievals from the OMI instrument. The retrieval quality of tropospheric column ozone has been improved. The presented methods are presented clearly and the paper is generally well written.

.→ We appreciate the useful comments, and tried to improve the manuscript, in accordance with reviewer's suggestion.

**General comments:**

C1. Could the authors also provide a comprehensive discussion of the limitations of the OMPROFOZ v2 algorithm and the potential sources of error in the ozone profile retrievals, and plans for the next version if there will be.

R1. The main objective of this paper is to describe the implementation changes for reprocessing OMI collection 4 ozone profile product. We have no plan for the next version with the fact that the OMI in-orbit operation is scheduled to be terminated soon. The OMPROFOZ v2.0 better represents the tropospheric ozone distribution with the less striping errors and the reduced spectral residuals (Fig. 12) and the UTLS profiles (Fig.13). As well, the seasonal and long-term consistency of the tropospheric ozone is improved compared to the previous version (Fig. 14). However, our product quality is still suffering from instrumental degradation, row anomalies, calibration errors, insufficient measurement information/a priori ozone dependence, and forward model errors. This paper includes the validation results with ozoensonde soundings at three EU stations because of the computational budget. When the reprocessed OMI dataset is available, OMI collection 4 product will be extensively assessed in the similar approach to be done for OMI collection 3 by Huang et al. (2017;2018) that evaluated the tropospheric and stratospheric ozone variables against global ozonesondes and MLS measurements with respect to the retrieval quality and long-term consistency. And then we will experiment OMI datasets to see how OMI ozone profiles could contribute to the trend analysis. We have revised the conclusion section to specify a validation plan for the upcoming version as follow:

"*In the follow-up paper to this work, the reprocessed OMI collection 4 ozone profile dataset will be thoroughly evaluated against a comprehensive dataset of ozonesonde soundings and MLS stratospheric ozone profiles for establishing geophysical validation results and for assuring the long-term consistency of OMI ozone profile product data quality*"

C2. Can you address the potential impact of cloud and aerosols on the accuracy of the ozone profile retrievals and is it possible to derive reliable near-surface ozone from UV measurements after significant improvements in the calibration and retrieval algorithm?

R2. The impact of cloud and aerosols on the spectral fitting is partly accounted for by fitting the cloud fraction and first-order wavelength dependent surface albedo. However, this treatment could be insufficient for thick clouds and heavy loading aerosols. These could be better simulated with the bi-directional reflectance distribution functions for the surface and the treatment of clouds as scattering layers, but which are not feasible in the operational use due to the computational budget. The spectral calibration and forward model calculations are of importance to determine the retrieval quality in the boundary layer. However, the measurement noises should be much improved. In this algorithm, we set 0.2 % in the UV 2 and 0.4 % in the UV1 for random-noise errors to stabilize the iterative fitting process.

C3. Better to summarize this analysis of the uncertainties associated with the improved algorithm updates and their impact on the accuracy of the ozone profile retrievals, maybe in a table.

R3. According to this comment, we have edited table 5 to provide a summary of comparison between PROFOZ v2.0 and ozonesonde.

Table 5. lists of ozoensonde stations[*] and comparison statistics[#] of the tropospheric column ozone between PROFOZ and ozonesondes

| Station | Hohenpeissenberg | Payerne | Uccle |
|---|---|---|---|
| Instrument | Brewer-Master | ECC[+] | ECC[+] |
| Country | Germany | Switzerland | Belgium |
| Lon, Lat (º) | 11.01, 47.3 | 6.57, 46.49 | 4.35, 50.80 |
| Elevation (km) | 0.98 | 0.49 | 0.10 |
| PROFOZ v1.0 | | | |
| No. of comparison pairs | 726 | 1025 | 893 |
| Mean Bias $\pm$ **1σ** (DU) | 4.20±7.38 DU | 2.22±6.85 DU | -0.74±6.08 DU |
| Mean Bias $\pm$ **1σ** (%) | 13.87±22.04% | 7.50 ± 19.78 % | -0.81±17.34 % |
| Correlation coefficient | 0.66 | 0.73 | 0.74 |
| PROFOZ v2.0 | | | |
| No. of comparison pairs | 815 | 1084 | 946 |
| Mean Bias $\pm$ 1σ (DU) | 3.30±5.95 DU | 0.99±5.15 DU | -2.09±5.12 DU |
| Mean Bias $\pm$ 1σ (%) | 9.94±16.52% | 2.87 ± 13.88 % | -5.11±13.05 % |
| Correlation coefficient | 0.81 | 0.85 | 0.83 |

[*]All data are downloaded from the World Ozone and Ultraviolet Data Center (WOUDC) data via http://www.woudc.org.
[+]Electrochemical concentration cell (ECC)
[#]The number of comparison pairs between OMI and ozonesonde for the tropospheric column ozone (900-200 hPa) during the period 2005 to 2020. Mean Biases and  1σ  standard deviations are in both DU (Dobson Unit) and % from (OMI− ozonesonde) × 100/ozonesonde.

**Specific comments:**

C1 Line 182-183: "three kinds of parameters are newly added to implement the slit function linearization and common mode correction as a pseudo absorber." Please clarify in this sentence if the slit function linearization parameters are also implemented as pseudo absorbers?

R2. According to this comment, we have clarified the slit function linearization parameters as follows:

"*three kinds of parameters are newly added to implement the slit function linearization (slit width coefficient, shape factor coefficient) and common mode correction as a pseudo absorber.*"

C2 Line 186: It's better to describe how the "covariance matrix" be constructed or refer to some references that has described it (May be Liu et at., 2010).

R3. According to this comment, we have clarified this by editing the associated sentence, "*They are assumed to be uncorrelated between fitting parameters, except for atmospheric profiles with a correlation length of 6 km, which gives* $\text{S}_a\ (i,j) = \sigma_i^a \sigma_j^a \exp(-|i-j|/6)$*, where  $\sigma^a$  is  with i and j being layer numbers.*"

C3 Line 237: I guess *Ie* represents the measured random noise errors.

R4. According to this comment, we have revised the one from *"measured errors"* to *"measured random noise errors"* for clarification.

C4 Line 283: May be add a plot of information contents or averaging kernel in Figure 2 for the layer of tropopause helps understanding the sentence "In the subtropical region, LLM may also provide incorrect information in the presence of high tropopause height…"

R4. Figure 2 clearly shows that LLM a priori provides the inconsistent information on tropospheric ozone in the latitude above 30ºN where the tropopause height is higher, compared to that taken from TB a priori. That is because that the LLM ozone climatology is only able to represent the ozone profile shape constrained with the lower tropopause height in the latitude above 30ºN during the winter. As a result, the LLM based TCO gives the abnormally high ozone features in the northern Europe area, consistently with the LLM a priori, implying the dependence of retrievals on a priori (less measurement information content). On the other hand, LLM based TCO gives independent ozone distribution in the subtropical area in spite that LLM a priori also gives the abnormally high ozone features, implying the existence of more independent measurement information. We think that Figure 2 is sufficient to address the importance of the a priori information on our OMI ozone retrievals and improvements of a priori data/PROFOZ v2.0 ozone data. The following figure illustrates the zonal mean DFS profile. This indicates the existence of a relatively larger measurement information over the upper troposphere in the subtropical area (25-30 N) than in the mid/high latitude. However, we decided not to include this figure in the revised manuscript for conciseness and just clarify it in this section.

[Figure]

Same as Fig. 2, but for mean Degrees of Freedoms (DFS) for Signal at each layer in the troposphere and lower troposphere. The dashed black line represents the tropopause.

C5 Line 357-359: How does the monthly averaged irradiance spectrum cancel out the common degradation existing in radiance and irradiance?

R5. Our retrieval prefers to the use of normalized radiance (radiance/irradiance) rather than radiance itself with the benefit of removing the extraterrestrial absorption signatures and canceling out the calibration uncertainties commonly existing in radiance and irradiance. As commented by the first reviewer (C8). optical elements are similar for radiance and irradiances, except for diffuser and folding mirrors. Actually, the solar diffuser degradation is one of main sources to degradation errors to irradiance measurements. Therefore, we revised from *"cancel out the common degradation existing in radiance and irradiance"* to *"to address seasonal variations of instrument characteristics that are common in both radiance and irradiance measurements"*. And we also prefer to the use of the monthly averaged irradiance than daily irradiance with the benefit of reducing the short-term noises in individual irradiance measurements, which improves out RMS of fitting residuals for both collection 4 (0.15→0.14) and collection 3 (0.24→0.17) as shown in Figure 7.

C6 In section 3.6, how about the correlation between slit functions parameters and ozone parameters in the retrieval?

R6. The correlation is ~ 0.01 for UV1 parameters, but increase 0.1 between UV2 slit width coefficient and the tropospheric ozone and 0.2-0.3 between the UV2 shape factor coefficient and the tropospheric ozone. We have pointed out this information in the revised manuscript as follows:

"*These PA coefficients are weakly correlated with ozone variables, except for the UV2 shape factor coefficient ($\Delta$k) (~0.2~0.3).*"

C7 In section 3.9, does the common mode correction improve the ozone profile retrieval other than just improving the fitting residual?

R7. The common model correction is implemented to address the remaining spectral residuals after soft calibration and hence to improve the fitting quality. In addition, this correction helps to smooth out the cross-track dependent biases in the tropospheric ozone retrievals, in particular at the extreme nadir-off pixels (please take a look at Fig. 8 d-f vs Fig.12 a-c).

---

## Author Comment (AC2)

**Overall Comments:**

I find that this is a carefully crafted paper, which I appreciate. I can tell the authors put a lot of effort into error assessment and to describing their algorithm improvements. They describe a v2 product that is a significant improvement over an already good performance by the v1 product. However, I believe the authors vastly understate the importance of their soft calibration approach in the success of the OMPROFOZ product. Nor do the authors adequately discuss the information content of their product given their soft calibration approach. The derived ozone profiles are effectively normalized to equatorial MLS profiles once per year. Therefore, the OMPROFOZ product is one that describes extra-tropical ozone profile variability and intra-annual ozone variability relative to MLS ozone profiles. There is nothing wrong with this approach, nor is it even new to this version. A product such as OMPROFOZ that better describes UTLS ozone variability is quite valuable. But the authors should clearly state up front (i.e. in the abstract or in the conclusions) what this product is and what it is not. In particular, it is not a product that independently measures long-term changes in ozone profiles. Their soft calibration approach is central to the definition of this product, yet the authors treat it almost as an afterthought with little mention throughout this paper. One gets the impression they themselves do not fully appreciate the role this normalization plays.

.→ We appreciate the sincerely comments, and tried to improve the manuscript, in accordance with reviewer's suggestion.

**Specific Comments:**

C1. Section 1, Line 51 Remove "been"    C2. Section 1, Line 56 Insert "to" before "evaluate"

R1 & R2 Accepted.

C3. Section 1, Line 66 Since this is the first reference to MLS in this paper, please indicate that you refer to the AURA instrument.

R3. The manuscript has been edited to accept this comment; "*Ten-years of the OMPROFOZ product were assessed in-depth in Huang et al. (2018;2017) through the spatiotemporal validation using global reference dataset collected from balloon-borne ozonesondes and space-borne Microwave Limb Sounder (MLS), which is one of the payloads onboard the Aura satellite, along with the OMI instrument.*"

C4. Section 2.1, Line 122 In the interests of full disclosure the authors should inform the readers that the row anomaly affects ALL of the UV1 channel. There are no rows that are reliably free of the effects of the anomaly, though longer wavelengths tend to be less effected than shorter ones, and lower rows less than higher ones.

R4. Thanks to specify the row anomaly issue in detail. The row anomaly is a well-known issue. The anomaly affected measurements are uncorrectable. In this section, we need to address which flagging are applied, rather than the detailed explanation of the anomaly behavior for assuring the conciseness and consistency of the context. Therefore, Schenkeveld et al. (2017) is referenced for the detailed behavior of the row anomaly in the revised manuscript.

C5. Section heading 2.1 is duplicated. Section 2.2 is missing.

R5. The number of the section has been corrected.

C6. Section 2.2, Lines 171-175 The authors state that the measurement noise reported in the OMI L1B product underestimates the true noise. This is a well-known problem across multiple instruments. It occurs because pseudo-random systematic errors, either in the measurements or in the model, are much larger than detector noise. The authors go on to state that their assumed minimum errors are uncorrelated between wavelength, but this seems to be a rather poor assumption. Many modeling and measurement errors are spectrally correlated, e.g. cloud modeling errors. Can the authors comment on the effect such an assumption has on their product?

R6. We agreed with what this reviewer addressed about the assumption on the measurement noises. The detector can be interfered between adjacent pixels. However, there is no data source about the information on the correlation behavior between wavelengths for OMI and other similar instruments. It is common to construct the measurement error covariance as a diagonal matrix, where measurement errors are assumed to be uncorrelated among wavelengths, as suggested by Rodgers (2000) who introduced an inverse theory based on an optimal estimation. There is no literature about the BUV ozone profile retrievals in which the spectral correlation of measurement errors is applied, based on our knowledge.

C7. Section 3.2, Line 298 Begin this sentence with, "In Version 1 these meteorological variables ..."

R7. Accepted.

C8. Section 3.5, Line 358 Unlike other BUV instruments, normalizing by the OMI measured solar irradiance does not reduce optical degradation errors. With OMI there are several optical elements not shared in common between Earth radiance measurements and solar irradiance measurements. In the case of solar irradiance these elements (diffuser and folding mirror) represent the primary sources of degradation. As a consequence, the Coll. 4 degradation corrections for radiance and for irradiance are completely separate. Furthermore, the degradation correction for irradiance measurements was derived by assuming constant solar irradiance over the mission (Figure 5c demonstrates this point). This is a reasonably good calibration approach for wavelengths longer than 300 nm, but not for the UV1 channel. There are clearly benefits to normalizing OMI radiances with a time-dependent solar irradiance, but cancellation of long-term optical degradation is not one of them. Having said all this, it's not clear why the authors are concerned about optical degradation given their soft calibration approach outlined in Section 3.8. Perhaps the authors should make it clear in Section 3.5 that their interest in improved solar irradiance measurements is solely to address seasonal variations in instrument calibration.

R8. Thanks for specifying the calibration characteristics for OMI. In previous version, the data processing uses the climatological solar spectrum derived from three years of daily OMI Level 1B product (2005-2007), which is switched to the monthly solar irradiances in the new version. with . As addressed by this review, applying the monthly solar irradiances does not reduce the degradation errors, but offers several benefits in reducing short-term noises and in addressing the seasonal variations of instrument characteristics that are common in radiance and irradiance measurements, compared to applying either climatological spectrum or daily spectrum. As shown in Fig.7.b, the fitting quality is improved due to switching from daily irradiances to monthly mean irradiances due to improved signal to noise ratio after averaging. According to this comment, we have revised the manuscript to address the seasonal variations in instrument calibration, with the improved solar irradiance spectrum as follows.

"*In order to reduce the short-term noise of individual measurements, the earlier algorithm implemented the use of climatological solar spectra derived from three years of OMI collection 3 irradiances (2005-2007). In the newer algorithm, collection 4 irradiances are tabled as monthly averages to address seasonal variations of instrument characteristics that are common in both radiance and irradiance measurements.*"

C9. Section 3.1, Lines 369 - 371.What does "implementations are identically applied" mean? If you mean to say that Coll. 3 and Coll. 4 data are treated the same for this experiment, please say so.

R9. As described in the original manuscript, Figure 7 shows the retrieval experiment to see the impact on the spectral fitting residuals due to switching OMI L1b data from collection 3 to 4. The radiance is mostly unchanged and so this comparison represents the impact due to switching irradiance. We stated that "In this experiment, the v2 implementations are identically applied without radiometric corrections (soft calibration and common mode correction are commonly turned off)". It means that both ozone profiles are retrieved with the v2 algorithm, but soft calibration and common mode correction are commonly turned off.

C10. Section 3.6 The authors could instill confidence in their pseudo-absorber approach to scene inhomogeneity if they were to demonstrate that their empirical parameters correlate with scene reflectivity changes or the small pixel column results contained in the OMI Level 1B product. The largest slit function errors will occur in the along-track direction at cloud edges, which can be identified via scene reflectivity or the small-pixel data. However, if such a comparison has already been shown in the referenced paper the authors may ignore this comment.

R10. We appreciate this important point. In the reference paper (Bak et al., 2019b), we already showed the correlation between pseudo-absorber coefficients ( $\triangle w, \triangle k$ ) and scene reflectivity changes as well as the improved fitting residuals with their pseudo-absorbers being implemented.

C11. Section 3.8 Per the authors' description in lines 454-463, OMI calibration is adjusted so that the retrieved ozone profiles match MLS + LLM profiles in the tropics. Per the description provided, this normalization occurs during the northern summer every year. While such an approach should help deal with systematic biases caused by the row anomaly, a once-per-year correction is inadequate to deal with the variable nature of the row anomaly. The authors should address the question of what intra-annual TCO errors may remain after the once-per-year soft calibration corrections.

R11. We would like to clarify the soft calibration. The soft calibration is intended to correct the slowly varying systematic biases, not to correct the rapidly varying biases. Therefore, soft calibration is composited of the spectral residuals between measurements and simulation, where the forward model simulation could be as accurate as possible, such as the summer tropics at nearly clear sky conditions. The ozone profile input is one of the main sources of errors in the forward model calculation and thereby the summer tropics is targeted with the fact that the daily ozone variability is relatively small. We prepare ozone profile inputs by merging MLS in the stratosphere and LLM in the troposphere, with the profile shape adjustment based on OMI column ozone. The soft calibration is an empirical correction and therefore they should be carefully derived and carefully applied. Therefore, we derived the soft spectra only as a function of instrument dimensions for each year, assuming that the soft spectra are representative of the slowly changed degradation errors, and implemented the soft calibration to correct the daily measured spectrum by interpolating the yearly given soft spectra. Both soft calibration and common mode correction either cannot account for the remaining calibration errors or induce the artificial errors somewhere. These could be the answer for some intra-annual TCO errors. However, Figure 14 still demonstrates the improvement of the seasonal bias with the update to PROFOZ version 2.

C12. Section 3.9 The authors should attempt to provide a physical explanation for the observed intra-orbital variations in residuals. Without a reasonable explanation how can the authors or the readers be confident a static CMC correction is appropriate and adequate? The most likely explanation for the observed residual variation is additive errors (e.g. stray light) and the row anomaly. Will the CMC as the authors have implemented it address errors introduced by stray light and the row anomaly?

R12. As mentioned in the original manuscript, the intra orbital variations in residuals implies the existence of a spectral dependence of the radiometric calibration and detector sensitivity on the signal represented by solar zenith angle, which is not completely treated by the soft calibration dependent only on CCD dimension. As mentioned in this comment, the solar zenith angle dependent features could be related to the straylight. It is assumed that the row anomaly contaminated pixels are screened out in this retrieval experiment. We observed the consistent component of fitting residuals, with the dependence on the solar zenith angle (Figure 14 d/e/f). Therefore, we carefully composited the common mode spectrum and then fitted them as a pseudo absorber. Comparing Figure 14 d-f and Figure 14 g-I demonstrated the improvement of the spectral fitting quality and comparing Figure 8.f and Figure 12.c also confirmed the improvement of the retrieval quality at cross-track positions 1-5.

C13. Figure 13 The MB and Std. Dev. labels appear to be reversed.

R13. Corrected.

C14. Figures 13 & 14 Given that the OMPROFOZ product is tied via soft calibration to MLS+LLM, it will be helpful to show readers similar comparisons of MLS+LLM to the same ozonesonde measurements (or at least measurements from the same stations). This may provide insight into how much of the observed TCO-sonde difference arises from the choice of soft calibration.

R14. As addressed in replying to comment 11, our soft spectra are derived as systematic biases due to slowly varying radiometric calibrations, not due to the forward model calculations in which MLS + LLM is used as an input once per year. Therefore, our ozone profile is not tied much to MLS + LLM. This paper aims for describing the algorithmic updates and its verification in details. In this study, the validation is limited at three ozonesonde stations due to the computation limit in the local machine. The PROFOZ v2.0 processor is being tested in the NASA OMI operational facility for reprocessing the entire mission. When the new product is available, we are planning to perform the global and long-term validation activities for both tropospheric and stratospheric ozone retrievals against ozonesonde and MLS measurements.

C15. Section 4.0, Line 583 The authors should avoid referring to the flags as "TOMS-based" and instead continue to reference the OMUANC product as the source of these flags.

R15. We have revised the manuscript to accept this comment as follows: "Those spikes could be attributed to row anomaly-contaminated retrievals unscreened with the row anomaly flags taken from OMI collection 3 L1b product. The related improvements in OMPROFOZ v2 retrievals are contributed by applying the stricter flags taken from the OMUANC product."

C16. Section 5.0, Lines 628-630 This brief mention of the soft calibration understates the role it plays in the performance of this product. The text suggests that its role is to merely keep the simulations close to the measurements, perhaps to keep them in a more linear regime. Surely, in an iterative retrieval algorithm that typically requires no more than 2-3 iterations (Section 2.2) dependence on the initial guess is not strong. The authors should acknowledge that the primary role played by their soft calibration is to eliminate the long-term drift observed in v1, and to remove some of the static and slowly varying row anomaly errors that have hitherto stymied all other attempts at retrieving ozone profiles from OMI data.

R16. The lines 628-630 provide a summary of the implementation changes from v1 to v2. Therefore, we stated the update to the soft calibration as "*The empirical soft calibration spectra are re-derived annually to be consistent with the updated implementations to remove the systematic differences between measured and simulated radiances.*". In both v1 and v2, the soft calibration plays an important role in eliminating the CCD-dependent systematic errors, in particular for reducing the stripes on the tropospheric ozone retrievals and the spectral fitting residuals. The main update to soft

calibration (v1→2) is expanding the dimension of calibration spectra to account for the degradation error. Actually, this work was initially experimented in the frame of collection 3 and the soft calibration played the primary role for eliminating the long-term drift. However, this role is reduced in the frame of collection 4. Therefore, the soft calibration plays a role in accounting for the remaining degradation errors.

C17. Section 5.0, Lines 643-648 The authors imply that the improved long-term drift is somehow related to switching from Coll. 3 to Coll. 4 and to some unidentified implementation details. Given the soft calibration approach in v2 it is unlikely that the improved calibration in Coll. 4 plays any role in this.

R17. The major updates from collection 3 to collection 4 is mostly related with the solar irradiance degradation correction as well as the improved quality flagging. The soft calibration is employed for accounting for the systematic biases of the normalized radiance (radiance/irradiance) as a function of CCD dimensions for each year. As indicated by this reviewer, the dependence of the data quality (PROFOZ v1 → v2) on the soft calibration is reduced with respect to the long-term consistency, due to the improvement of L1b data quality (collection 3→4). However, the temporal dependent soft calibration plays an important role in improving the data quality and consistency. Therefore, we decided to implement the yearly soft calibration in the v2 implementation. Please take a look at the following figure, comparing the retrieval results with yearly soft calibration and the 2005 soft calibration, respectively.

[Figure]

Same as Figure 12 in the manuscript. (a, d) Spectral fitting residuals (%) averaged in the latitude of 60°S and 60°N from OMI measurements on 15 June 2018, (b,e) the global distribution of tropospheric column ozone (TCO, DU), and (c,f) anomalies of TCO as a function of 18 latitude bands, but for comparing the retrieval results on 2018m0615 with yearly soft calibration and the 2005 soft calibration, respectively Note, the 2005 soft calibration represents the correction spectrum derived from measurements on 2005m0711-17.

---

## Author Comment (AC3)

**Overall Comments:**

In this work, Bak et al. provide a thorough description of "An improved OMI ozone profile research product version 2.0 with collection 4 L1b data and algorithm updates". Although it is of significant scientific value to the space-borne atmospheric monitoring community, its presentation can be substantially improved, both in terms of research results and general phrasing.

.→ We appreciate the useful comments, and tried to improve the manuscript, in accordance with reviewer's suggestion.

**Specific Comments:**

C1 Line 59: A degradation is provided in %, but the time range is not specified. Is this per decade, or 'now' with respect to beginning of mission? Idem for the wavelength stability in nm (also in line 155).

R1. The magnitudes of degradation and wavelength shift were taken from Schenkeveld et al. (2017) where OMI long-term stability is assessed over the period of 2005 to 2014. In this revised manuscript, Kleipool et al. (2022) has been referenced to indicate the long-term stability of OMI instrument over the period of 2005 to 2021. Accordingly, the related sentence has been revised as "the OMI instrument show progressively low optical degradation over the mission, with a change of ~ 3 % in the radiance over roughly 1.5 decades (Kleipool et al., 2022)" in introduction (Line 59 in the original manuscript). The corresponding statement in line 155 was removed in the revised manuscript for assuring the conciseness and consistency of the context.

C2 Line 62: "satellite ozone profile products have not been reliable for long-term analysis" sounds too strong. This should be rephrased.

R2. We agreed with this comment. It was rephrased as "OMI instrument show progressively low optical degradation over the mission, with a change of ~ 3 % in the radiance over roughly 1.5 decades (Kleipool et al., 2022). However, the long-term reliability of the OMPROFOZ product, particularly concerning tropospheric ozone measurements, remains susceptible to optical instrument degradation (Gaudel et al., 2018; Huang et al., 2018, 2017).."

C3 Line 68: "latitude/season/viewing geometries" can be combinedly referred to with an optical path dependence? Please discuss appropriately.

R3. According to these comments, "and the dependence of retrieval quality on the latitude/season/viewing geometries" has been revised as "and suggested the need to address the spatiotemporal variations of the retrieval performance and the related cross-track dependency".

C4 Lines 84-85: "Note that OMI measurements have been reprocessed to deliver the new collection 4 dataset" Does this apply to the entire time series, or a limited period only?

R4 OMI collection 4 L1b data is available for entire time series
    (https://disc.gsfc.nasa.gov/datasets/OML1BRUG_004/).

C5 Lines 109-110: It is not clear where the codes between brackets refer to.

R5 The code in bracket indicates the DOI for the given dataset, which has been deleted in the manuscript, but provided as a part of the reference (e.g., Kleipool, Q. (2021a), OMI/Aura Level 1B Averaged Solar

Irradiances V004, Greenbelt, MD, USA, Goddard Earth Sciences Data and Information Services Center (GES DISC), Accessed: [2023-07-21], 10.5067/Aura/OMI/DATA1401, 2021a)

C6 Lines 126-127: Does the "KNMI flag" name apply in general, or in this work only?

R6 Most of the information regarding the OMI row anomaly is sourced from Schenkeveld et al (2017). The referenced paper also uses "KNMI/NASA (row-anomaly) flag", NASA and KNMI flagging results to refer to the row-anomaly flag derived with each detection method.

C7 Lines 135-136: "In addition, OMI total column ozone product (OMTO3d) is used in deriving empirical correction spectra." This sounds too vague. A reference or more detailed explanation is needed.

R7. According to this comment, we provided more details in the revised manuscript, as like "We applied OMI total column ozone product (OMTO3d) to adjust the ozone profile shape used as an input for deriving empirical correction spectra (Sect. 3.8)."

C8 Line 149: "48 x 26" instead of "38 x 26"?

R8 Thanks. It is corrected.

C9 Figure 1 may be too detailed or might need an explanation of the variables used.

R9. The most of the variables have been described in the text after Figure 1. In the revised manuscript, we have mentioned more variables and described it in more detail.

C10 Line 175: "assuming that measurement errors are uncorrelated among wavelengths" Is this a valid / common approach then? Please specify or provide referencing.

R10. It is very common to construct Sy as a diagonal matrix, where measurement errors are assumed to be uncorrelated among wavelengths, as suggested by Rodgers (2000) who introduced an inverse theory based on an optimal estimation.

C11 Lines 186-187: Explain with reference(s) how the correlation length is applied in the covariance matrix.

R11. We have specified how to apply the correlation length in the revised manuscript as "They are assumed to be uncorrelated between fitting parameters, except for atmospheric profiles with a correlation length of 6 km, which gives $\mathbf{S}_y\,(i,j) = \sigma_i^a\sigma_j^a\exp(-|i-j|/6)$, where $\sigma^a$ is a priori error, with i and j being layer numbers"

C12 Lines 207-209: What happens to levels below the effective surface pressure?

R12 As mentioned, the bottommost level is the surface pressure.

C13. Line 276: Where is this tropopause pressure obtained from (also in Figure 2)?

R13. In this figure, the meteorological variables including the tropopause pressure are commonly taken from the NCEP reanalysis data, only to see the impact of applying different a priori ozone profile information on the retrieval. For clarification, we have inserted "It is noted that the meteorological variables are commonly taken from the NCEP reanalysis data to see the impact of applying different *A* priori ozone data on the retrieval." in the caption of Figure 2.

C14 Lines 303-305: "However, the data transmission has been accidently halted since June 2011 and hence climatological monthly mean data have been used as a back-up in the data processing." Not clear whether this applies to v1 or v2.

R14 As mentioned, this happened to the v1 data processing. To avoid this risk, the meteorological input is switched to the internal meteorological products (OMUFPSLV, OMUFPMET)

C15 Lines 327-329: "BW measurements were better parameterized as quadratic temperature-dependent coefficients with uncertainties of 0.25-2 % whereas for BDM measurements fitting residuals of 2-20 % remains." Also, lines 346-347 "radiometric uncertainties of the new reference spectrum are below ~ 1 % whereas for SAO2010 those range from 5% in the longer UV part to 15 %" That's two times a factor of ten improvement, and might be the most impactful changes. This might be stressed in the conclusions and/or better explained and referenced?

R15. Note that the factor of 10 improvement for cross section occurs in the cross section fitting residuals after the temperature-dependent parameterization. It can also have a large impact on ozone profile retrievals when soft calibration is turned off (Bak et al., 2020). With soft calibration derived using consistent cross sections, some of the systematic differences due to cross sections can be greatly reduced; using BW can still improve the retrievals due to its better temperature dependence, but it does not cause the most impactful changes. The high-resolution solar reference is used mostly for wavelength and slit function calibration. Despite the much better agreement of TSIS HSRS with TSIS SIM observation than the SAO2010 reference, the improvement on the ozone profile retrieval is also relatively smaller, changing the tropospheric layer ozone columns by 5-7% (Bak et al., 2022).

C16 Why are different temperatures chosen for plots (a) and (b) in Figure 4?

R16 We plotted each cross-section data at all temperatures available from the original database ("not chosen"). In the original manuscript, it has been noted as "the BW dataset provides improved temperature coverage from 193 K to 293 K, every 20 K over the BDM dataset given only at five temperatures above 218 K"

C17 Line 401: Explain cross-product and $I_h$ in the equation.

R17 Each represents "spectral convolution" and high-resolution radiance spectrum. In revised manuscript, the equation has been simply expressed as $\frac{\partial I}{\partial p}$ (p = $w$ $or$ $k$) for conciseness and clarify. The implementation details to get $\frac{\partial I}{\partial p}$ ($\frac{\partial S}{\partial p} \otimes I_h$) through the slit function linearization can be found in Bak et al. (2019b), as indicated in the end of the corresponding section 3.6.

C18 Lines 411-412: A factor $f_c$ seems to be missing the second term of the formula (while on the other hand $f_c$ is not explained).

R18 We have specified the indicated equation as "I = I ($R_{sfc}$, $P_{sfc}$)(1 − $f_c$) + I ($R_{cloud}$, $P_{cloud}$) $f_c$ where R and $P$ represent reflectivity and pressure at bottom level (surface or cloud) with $f_c$ as an effective cloud fraction"

C19 Lines 412-413: "finer grids than at least 4 pixels per FWHM so that the spectral convolution is applied to account for OMI spectral resolution" This is too brief to be clear.

R19 To address this comment, we have revised this sentence as "According to the Nyquist criterion (Goldman, 1953), individual spectra need to be simulated at grid spacings finer than a minimum of two pixels (four pixels in practice) per spectral resolution"

C20 Table 4 and Table 5 seem to be missing?!

R20 We are very sorry to make this big mistake. Tables 4 and 5 have been inserted in Section 3.7 and Section 4, respectively.

C21 Lines 428-431: this is unclear if half-streams are not explained.

R21 The half-streams are just discrete ordinate streams in the half-space. For clarification, the definition of the half streams has been specified in the revised manuscript.

C22 Line 433: "(2 vs. 6) and number of layers (24 vs. 72)" which versions are compared here?

R22. As mentioned in the original manuscript (a look-up table (LUT)-based correction is performed, which enables to adjust the differences in RT variables due to different number of streams (2 vs. 6) and number of layers (24 vs. 72) as well as neglecting polarization effect), a LUT based correction is applied to improve the approximation errors in radiative transfer calculations, arising from low number of streams and coarse vertical layering as well as neglecting polarization effect. That is, the operational algorithm runs on-line RT model in scaler mode with the atmospheric layers of 24 and two streams. This approximation error is compensated with the look-up table correction in which the RT variables are pre-calculated in vector with the atmosphere layers of 72 and six streams.

C23 Lines 435-438: "Note that the Ring simulation remains unchanged from v1 algorithm; the spectral structure of the Ring signal is externally simulated with the iterative fitting of amplitude of the Ring spectrum and then subtracted from the measured spectral reflectance." Please provide reference.

R23 This concept of accounting for ring effect has been adopted from the OMPROFOZ v1.0 algorithm so that Liu et al. (2010) is cited as a reference according to this comment.

C24 Lines 446-447: "along-track stripes that are commonly found in OMI trace gas products" Should this be clear from the plots? Please provide references.

R24 The cross-track dependent biases are clearly found in both fitting residuals (Figure 8.a) and tropospheric ozone retrievals (Fig 8. b and c) (along-track stripes in the global map). We thus derive the cross-track dependent soft calibration spectra (Fig. 9) to reduce stripes (see. Fig. 8 d-e-f). According to this comment, we have added several references (e.g. Wang et al. 2016; Kroon et al., 2011; LN Lamsal et al. 2021)

C25 Section 3.9: Please explain important "pseudo-absorber" concept for clarity of common mode correction.

R25 The common mode correction is implemented to address the remaining spectral residuals after soft calibration. The soft calibration is applied to the measured spectrum once before ozone spectral fitting. The common model spectrum is iteratively adjusted as a pseudo-absorber to absorb the remaining residuals. This common mode correction is intended for improving the fitting accuracy, but also helps to smooth out the cross-track dependent biases in the tropospheric ozone retrievals, in particular at the extreme nadir-off pixels (please take a look at Fig. 8 d-f vs Fig.12 a-c).

C26 Section 4: Why the severe limitation of having 3 sites above Central Europe only? To what extent are these results globally valid?

R26 The main objective of this paper is to describe the implementation changes for re-processing OMI collection 4 ozone profile product, with algorithmic updates including radiative transfer calculations,

slit function parameterization, soft calibration, new pseudo absorbers and input changes (a priori ozone data, high-resolution solar reference, ozone cross-sections). And we verified the improvements of the implementation changes by evaluating the spectral fitting quality and the geophysical structures of the retrieved ozone distribution. Ozone profile retrievals were also evaluated in terms of quality and short/long-term consistency against the ozonesonde measurements achieved from three stations in the Central Europe. We limited this comparison with ozeonsondes at three stations due to the computation budget. However, we definitely have a plan to report the global and long-term validation results when the fully reprocessed product is available.

C27 Line 545: A 3 sigma for 'extreme values' seems quite low. Real outliers are typically beyond that.

R27 This comparison intended for changing the retrieval quality from v1 to v2. Therefore, we don't necessarily apply the tight outliers for a fair comparison. However, we definitely have a plan to report the global and long-term validation results when the fully reprocessed product is available. According to this comment, we will apply the tight outliers in the upcoming validation activity.

C28 Line 611: What does "continued externally" mean?

R28. The OMPROFOZ product is officially released just for the initial version and has never been changed, while each algorithm detail has been improved without applying into the operational algorithm. For clarification, we have revised the related phrase as follows "~which has not been updated since its initial data release. In this paper, we introduce algorithmic updates for reprocessing the OMPROFOZ product to enhance the retrieval accuracy and to ensure long-term consistency. This second version will be released at GES-DISC while the first version will remain archived at AVDC."

**Technical corrections**

C1 Line 51: "has been contributed"

R1 It has been edited as "has contributed"

C2 Line 113: "shorter" refers to "shorter-wavelength"?

R2 Yes. "the shorter spectra" has been edited to "the shorter wavelength"

C3 Line 132: "stricter and more reliable"

R3 Thanks. "stricter and reliable" has been edited to "stricter and more reliable"

C4 Line 133: "flags are raised" instead of "flags are flagged"

R4 Thanks. "flags are flagged" has been edited to "flags are raised"

C5 Please explain norm symbols (with lower and upper 2) in equation (2).

R5. "where $\| \quad \|_2^2$ denote the sum of each element squared" has been inserted after the equation 2.

C6 Line 471: "stripes"

R6. Accepted.

C7 Line 549: "relatively not serious" is unscientific phrasing.

R7 The indicated phrase has been edited as "the row anomaly affects the data in a few rows"

C8 Figure 13: The legends for mean bias and standard deviation seem to be reversed.

R8. Thanks for finding this mis-print. The legends have been corrected.

C9 Lines 650: please put dates on "in progress" for future reference.

R9 We have expected that the PROFOZ v2.0 data will be released in 2024, probably before July for the entire mission, but the schedule is changeable with the simulation in the data processing center.